# ClarifyVC: Clarifying Ambiguous Commands in Vehicle Control with a Hybrid Data Augmentation Pipeline

**Hange Zhou**[1,*]                **Zhonglin Jiang**[2,3,*]                **Yingjie Cui**[2,3,*]

**Mingzhe Zhang**[1]                **Xiaotang Wang**[1]                **Hengwei Dai**[1]

**Qiyao Yu**[4]                **Yong Chen**[3,†]                **Yongqi Zhang**[1,†]

[1] The Hong Kong University of Science and Technology (Guangzhou)
[2] Nanjing University of Information Science and Technology
[3] Geely Automobile Research Institute (Ningbo) Co., Ltd.
[4] The University of Hong Kong

## Abstract

Natural language interfaces for vehicle control must contend with vague commands, evolving dialogue context, and strict protocol constraints. We introduce **ClarifyVC**, a unified framework that integrates a hybrid data-augmentation pipeline (*ClarifyVC-Data*), reference models trained on the data (*ClarifyVC-Models*) and a evaluation protocol (*ClarifyVC-Eval*). The agent-orchestrated pipeline generates diverse, ambiguity-rich dialogues from real-world seeded queries under schema and safety constraints, while the evaluation protocol systematically probes single-turn parsing, conservative clarification under extreme fuzziness, and multi-turn grounding. Fine-tuning on ClarifyVC-Data yields consistent gains—up to 15% higher parsing accuracy, 20% stronger ambiguity resolution, and 98% protocol compliance—across realistic in-cabin scenarios, with human-in-the-loop assessments confirming high realism, coherence, and applicability. ClarifyVC thus advances beyond simulation-only datasets by tightly coupling real-world grounding with scalable generation and standardized evaluation, and provides a generalizable pipeline for broader interactive control domains. Our code and dataset are available at: https://anonymous.4open.science/r/ClarifyVC.

## 1 Introduction

Natural language interfaces are becoming a cornerstone of interactive control systems, from autonomous vehicles (Wen et al., 2024) to smart homes (Thukral et al., 2025), robotics (Sikorski et al., 2025), and other embodied agents (Bick et al., 2024). These systems require the ability to interpret vague instructions, maintain multi-turn dialogue context, and execute actions under strict protocol constraints. In the automotive domain, the rise of autonomous vehicles has already transformed human–machine interaction, making natural language commands crucial for intuitive and trustworthy control of hundreds of onboard functions (Zheng et al., 2024; Wang et al., 2024a). However, Vehicles face pervasive ambiguity—user commands are often vague, protocol mappings incomplete, and existing evaluation metrics inadequate (Ma et al., 2024). Traditional intent detection and slot-filling methods perform poorly under ambiguity and context shift, while current benchmarks lack realism, coverage, or failsafe metrics (Chun et al., 2025).

Public perception reflects these gaps: 58% of individuals feel uneasy about in-car voice assistants which is a key part of autonomous driving stack, and 25% express complete distrust in their relia-

---

*Equal contribution.
†Corresponding author. Email: yongqizhang@hkust-gz.edu.cn

bility (Wenskovitch et al., 2024; Peng & Shang, 2024). A core reason is that current LLMs, though strong in general reasoning, struggle in safety-critical control (Brahman et al., 2024). They hallucinate under ambiguous instructions, fail to request clarifications when uncertain, and lack strict protocol adherence in task orchestration (Dai et al., 2024). These weaknesses are compounded by the absence of high-quality, reality-grounded datasets and standardized evaluation protocols, limiting progress toward reliable in-vehicle dialogue systems (Nguyen et al., 2024; Zou et al., 2024). To close this gap, we introduce ClarifyVC, a unified framework for clarifying ambiguous commands in vehicle control. It integrates a hybrid data-augmentation pipeline (ClarifyVC-Data), reference models trained on the data (ClarifyVC-Models) and a three-tier evaluation protocol (ClarifyVC-Eval) to evaluate the data quality and model performance. Our work targets the safety-critical task of mapping natural-language in-car voice commands—often ambiguous or under-specified—into strictly validated, schema-aligned function calls, requiring accurate parsing, conservative ambiguity detection, and multi-turn parameter grounding.

At the core of ClarifyVC-Data is a hybrid augmentation pipeline seeded from over 20k authentic in-vehicle commands drawn from a proprietary corpus of 4M+ production-level interactions. Through structured ambiguity injection, adversarial perturbations, and multi-turn clarification, the pipeline synthesizes ambiguity-rich yet protocol-compliant samples that target robustness and safe execution. The resulting dataset, ClarifyVC-Data, has been validated through human evaluation and distributional alignment experiments, demonstrating close correspondence to real-world usage patterns. Fine-tuning LLMs on this data yields an average 15% improvement in parsing accuracy, underscoring the pipeline's practical value for safety-critical language interfaces.

Beyond dataset construction and training, we introduce ClarifyVC-Eval, a three-tier evaluation protocol that explicitly targets real-world ambiguity in function-call tasks (Jiang et al., 2024b; Chao et al., 2024; Wu et al., 2024; Jiang et al., 2024a). ClarifyVC-Eval plays a dual role: (i) it audits the benchmark itself—testing whether the data is realistic and ambiguity-rich—and (ii) it provides a unified lens to evaluate model capabilities in semantic parsing, execution fidelity, and safety compliance. By jointly assessing data validity and functional reliability, the protocol addresses a key gap in prior work, which typically isolates dataset realism from model accuracy and thus misses their interaction in safety-critical settings. To operationalize the data-side audit, we additionally define a *Dataset Quality Score (DQS)* that aggregates ambiguity diversity (AD), protocol compliance (PC), and realism (R). Together, ClarifyVC-Eval and DQS constitute a comprehensive, scalable framework for auditing datasets and benchmarking models under realistic ambiguity.

Extensive experiments demonstrate that ClarifyVC substantially improves performance in control tasks. Fine-tuned models achieve 15% higher parsing accuracy, 20% better ambiguity resolution, and 98% protocol compliance, while also reducing inference latency by 30% compared to baseline systems. Additional ablation studies confirm the necessity of each module in the data augmentation pipeline, yielding the best trade-off between diversity, coherence, and adherence. Multi-run evaluations further validate robustness, showing consistently low variance (<1%) and statistically significant improvements across metrics. Human-in-the-loop assessments corroborate these results, with expert annotators rating generated dialogues highly on realism, coherence, and practical applicability. Together, these findings highlight ClarifyVC as a reliable and efficient framework for robust language understanding under real-world ambiguity in vehicle control. In summary, our contributions are threefold:

1. **ClarifyVC Framework**: A unified framework for clarifying ambiguous commands in vehicle control and interactive systems. It integrates a hybrid data pipeline, a three-tier evaluation protocol, and reference models, offering an end-to-end standard for safe and deployable language interfaces.

2. **ClarifyVC-Data&Models**: A hybrid, reality-grounded, and human-validated dataset built from 20k+ real-world seed commands, expanded with controlled fuzziness and adversarial variants. By training on the high-quality data,we release reference models that show consistent gains in accuracy, clarification, and safety compliance.

3. **ClarifyVC-Eval**: A three-tier evaluation protocol that disentangles under-specification, ambiguity clarification, and multi-turn grounding, along with a Dataset Quality Score which ensures the benchmark aligns with real-world distributions and maintains high-quality standards. By explicitly targeting these failure families, the protocol enables comprehensive and safety-aware assessment of function-call understanding, addressing gaps left by conventional single-turn accuracy.

## 2 RELATED WORK

### 2.1 METHODS FOR CLARIFYING AMBIGUITY AND MULTI-TURN COMMAND PARSING

Natural–language command understanding has progressed from structure-aware parsers to end-to-end LLM solutions for mapping utterances to executable actions (Zheng et al., 2024; Wang et al., 2024a). Early pipelines emphasized schema-constrained intent/slot structures and hierarchical modeling (Sriram et al., 2019; Wang et al., 2024b; Okur et al., 2023). More recently, LLMs have been applied to enable direct intent grounding, rule translation, and task formalization (Shao et al., 2024; Choudhary et al., 2024; Manas & Paschke, 2023).

In the domain of vehicle or visual command understanding, datasets like Talk2Car (Deruyttere et al., 2019), CI-AVSR (Dai et al., 2022a), and doScenes (Roy et al., 2024) provide real-world instruction–action pairs and visual grounding contexts, but primarily support single-turn mapping rather than interactive clarification. Beyond single-turn parsing, logical disambiguation methods such as LogicalBeam (Bhaskar et al., 2023) have been explored, while other datasets and studies (e.g. CHAMBI) highlight cross-cultural or spatial ambiguity challenges (Zhang et al., 2024b; Saparina & Lapata, 2024). Frameworks for task decomposition and retrieval-augmented decision making further support complex instruction following (Shen et al., 2024; Yang et al., 2024). Parallel streams examine unimodal parsing (Zhang et al., 2024a), synthetic data generation (Liu et al., 2024), and distillation for instruction following (Ding et al., 2024), alongside domain-specific command datasets (Liu et al., 2023; Li et al., 2024). Together, these approaches offer modeling, training, and data-centric tools for tackling ambiguity and multi-turn semantics in command parsing.

### 2.2 LIMITATIONS OF EXISTING APPROACHES AND OUR POSITIONING

Despite steady progress, three gaps persist. (1) *Ambiguity management and uncertainty signaling.* Many systems lack explicit mechanisms to detect under-specification, trigger clarifying questions, or expose calibrated confidence, which is critical in safety-sensitive interaction (Pramanick et al., 2022; Wenskovitch et al., 2024; Lee et al., 2024). (2) *Evaluation scope.* Benchmarks often emphasize single-turn parsing, narrow modalities, or synthetic distributions (Zhang et al., 2024a; Liu et al., 2024; 2023; Li et al., 2024), offering limited coverage of multi-turn grounding and protocol-aware execution; even task-centric frameworks (Shen et al., 2024; Yang et al., 2024) provide only partial visibility into clarification behavior. (3) *Data realism and compliance.* Instruction-generation and distillation pipelines (Ding et al., 2024) rarely tie ambiguity to real logs or enforce function-call protocols, hindering transfer to deployed systems.

ClarifyVC addresses these gaps with a unified framework that: (i) couples a hybrid, real-log–seeded augmentation pipeline with controlled adversarial/fuzzy evolution to surface realistic ambiguity; (ii) introduces a three-tier evaluation protocol that disentangles single-turn parsing, clarification under extreme fuzziness, and multi-turn grounding with execution checks; and (iii) reports reference models trained on the data with protocol-aligned metrics. Our data generation pipeline, benchmark, and evaluation protocol demonstrate strong generalization and practical utility in real world settings. Beyond in-cabin voice control, the framework's clarification strategies and compliance-oriented evaluation naturally extend to broader domains of human-machine interaction, including smart homes, medical dialogue, and embodied intelligence, where safety and interpretability remain critical.

## 3 METHODOLOGY

We instantiate **ClarifyVC** as a unified framework comprising a hybrid data-augmentation pipeline (*ClarifyVC-Data*), reference models trained on the data (*ClarifyVC-Models*) and a safety-aware three-tier protocol (*ClarifyVC-Eval*). Seeded with 20k+ authentic in-vehicle commands (drawn from 4M+ production logs), the pipeline expands queries via structured ambiguity injection, adversarial perturbations, and multi-turn clarification under protocol constraints. The resulting corpus is validated through human studies and distributional alignment with real-world usage.

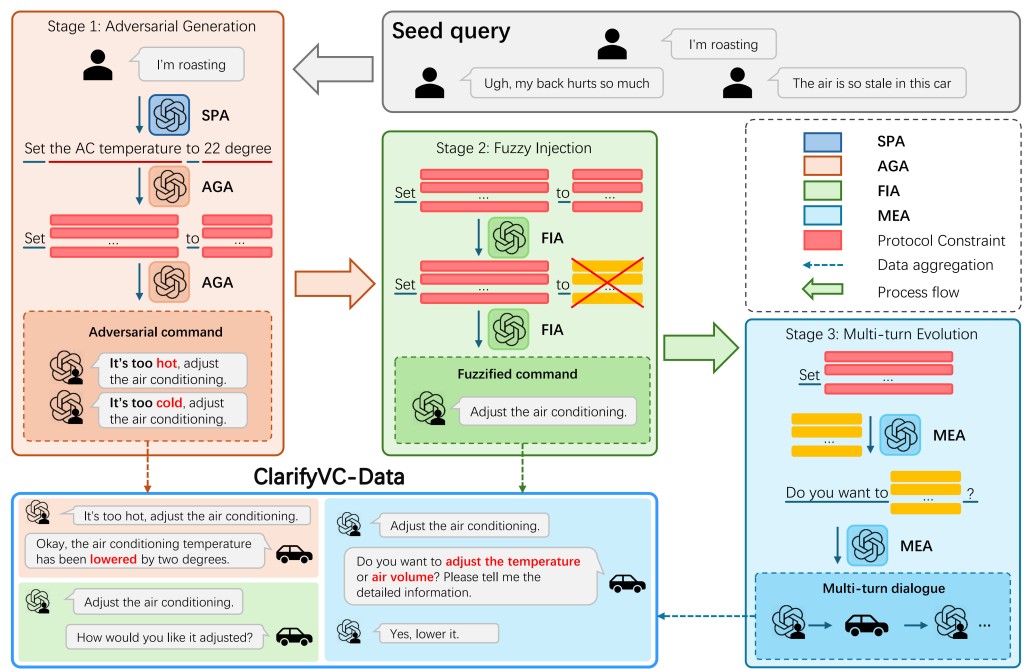

Figure 1: Agent-orchestrated, stage-wise generation flow. A schema-constrained pipeline executes semantic parsing, adversarial construction, fuzz injection, and multi-turn evolution on real-world–seeded commands to synthesize ambiguity-rich single- and multi-turn dialogues under protocol constraints. The resulting corpus forms ClarifyVC-Data: a hybrid, realism-aligned, human-validated benchmark with standardized function-call annotations and broad ambiguity coverage.

## 3.1 CLARIFYVC-DATA: AGENT-ORCHESTRATED PIPELINE

**Stage-wise pipeline.** We adopt an *agent-orchestrated* modular pipeline with four stages: Semantic Parsing Module (SPA), Adversarial Generation Module (AGA), Fuzz Injection Module (FIA), Multi-Turn Evolution Module (MEA), each implemented with prompt-engineered, pre-trained LLMs without task-specific fine-tuning. Specifically, SPA/FIA/MEA are implemented with *DeepSeek-R1* (API-based) for semantic parsing, fuzz injection, and multi-turn dialogue evolution, while AGA is realized with *Qwen2.5-72B* (via vLLM) to perform protocol-constrained adversarial rewriting. These choices combine scalability and strong instruction-following, while ensuring modularity for drop-in substitution. The total compute cost of synthesizing the benchmark remains modest, as generation relies primarily on API calls and lightweight orchestration:

- **SPA** parses each seed command into $(I, E, P)$ as standardized grounding.
- **AGA** produces syntactically valid yet ambiguous variants $c_{\text{adv}}$ under protocol constraints.
- **FIA** converts $c_{\text{adv}}$ into softer fuzzed instructions $c'$ (parameter omission, subjective modifiers, mild distortion); both tiers are retained.
- **MEA** expands $c'$ into coherent multi-turn dialogues $D$ for long-horizon grounding.

The adopted sequence (SPA $\rightarrow$ AGA $\rightarrow$ FIA $\rightarrow$ MEA) is not arbitrary but empirically validated. We conducted controlled ablations that permuted the order or removed individual stages. Evaluation results (Table 17) show that the default order achieves the best balance across ambiguity diversity, dialogue coherence, and protocol adherence. For example, reversing FIA and AGA substantially reduced diversity, while removing either stage markedly degraded ambiguity coverage. This confirms that the chosen order is optimal for synthesizing realistic yet challenging interactions.

This yields a hierarchical pool: (1) SPA+AGA $\Rightarrow c_{\text{adv}}$; (2) +FIA $\Rightarrow c'$; (3) +MEA $\Rightarrow D$. To encourage diversity while preserving operational validity, each sample is scored by

$$Q(c) = \alpha \cdot H(c) + (1 - \alpha) \cdot \mathbb{I}(c \text{ is protocol-compliant}), \quad \alpha = 0.6, \tag{1}$$

where $H(c)$ is ambiguity entropy and $\mathbb{I}(\cdot)$ indicates compliance (Appendix B.2).

**Reference models (ClarifyVC-Models).**  We obtain ClarifyVC-Models by supervised fine-tuning open-source backbones (e.g., LLaMA3-8B, Qwen2.5-7B/72B, DeepSeek-R1-Distilled) on ClarifyVC-Data with schema-aligned function-call targets, using a teacher-forced cross-entropy objective and JSON-schema–constrained decoding at inference(Experiment settings can be seen in Appendix B). Training is performed with early stop on a delayed test split and evaluated on a separate 2k test set, averaged on 5 random seeds (std.,$< 1\%$). We release the **Qwen2.5-7B-SFT** checkpoint and training/evaluation configs. Notably, while larger backbones (14B, 32B, 72B) show strong results, the 7B model achieves the best trade-off between accuracy and computational efficiency, reducing inference cost by an order of magnitude while delivering comparable or superior performance under our protocol. As will be shown in Table 2, the models fine-tuned with ClarifyVC-Data consistently surpass the zero-shot base models in different scenarios.

## 3.2 CLARIFYVC-EVAL: EVALUATING DATASET QUALITY AND MODEL PERFORMANCE

In this part, we introduce ClarifyVC-Eval to evaluate the quality of ClarifyVC-Data and ClarifyVC-Models. A Dataset Quality Score (DQA) is used to ensure the benchmark aligns with real-world distributions and maintains high-quality standards. Meanwhile, a three-tier evaluation protocol that disentangles under-specification, ambiguity clarification, and multi-turn grounding, is proposed to evaluate the generation quality of different LLMs.

**Automated quality and human validation for Dataset.**  We summarize dataset quality using the Dataset Quality Score (DQS):

$$\text{DQS} = \lambda_1 \cdot \text{AD} + \lambda_2 \cdot \text{PC} + \lambda_3 \cdot \text{R}, \quad (\lambda_1, \lambda_2, \lambda_3) = (0.4, 0.3, 0.3), \tag{2}$$

where AD, PC, and R are three complementary measures capturing ambiguity coverage, schema correctness, and realism of recovery pathways. We select $(0.4, 0.3, 0.3)$ via grid search to maximize Spearman correlation with human ratings while preserving consistent ranking across baselines. Full sweeps are provided in Appendix C.4.

The exact definitions of AD, PC and R are provided below.

**Ambiguity Diversity (AD).**  AD quantifies how well the dataset covers the five major fuzzy categories (intensity, boundary, entity, mode, referential). Let $\mathcal{A}$ be the set of ambiguity types and $p(a)$ the empirical proportion of type $a \in \mathcal{A}$. We compute:

$$\text{AD} = 1 - \frac{\text{KL}(p(a) \,\|\, u(a))}{\log |\mathcal{A}|}, \tag{3}$$

where $u(a)$ is the uniform target distribution. Higher AD indicates more balanced ambiguity patterns, avoiding collapsed or single-type dominance.

**Protocol Compliance (PC).**  PC measures the proportion of samples whose gold function calls satisfy the production HMI schema and rule-engine constraints:

$$\text{PC} = \frac{1}{N} \sum_{i=1}^{N} \mathbb{I}\big[c_i^* \in \mathcal{S}_{\text{schema}} \ \wedge \ c_i^* \in \mathcal{S}_{\text{safety}}\big], \tag{4}$$

where $\mathcal{S}_{\text{schema}}$ denotes JSON action–slot validity and $\mathcal{S}_{\text{safety}}$ captures type, range, and dependency rules. PC reflects structural correctness of the dataset's functional targets.

**Realism (R).**  R measures the alignment of each sample with real in-cabin interaction patterns. For each instruction $x_i$, we retrieve the $k$ most similar real-log utterances $\{x_i^{(j)}\}$ (using embedding similarity) and check whether its gold call $c_i^*$ matches the dominant action–slot pattern in retrieved logs:

$$\text{R} = \frac{1}{N} \sum_{i=1}^{N} \mathbb{I}\Big[c_i^* \in \text{Mode}\big(\{c(x_i^{(j)})\}_{j=1}^{k}\big)\Big]. \tag{5}$$

R therefore evaluates whether dataset samples resemble historically observed user behaviors rather than synthetic artifacts.

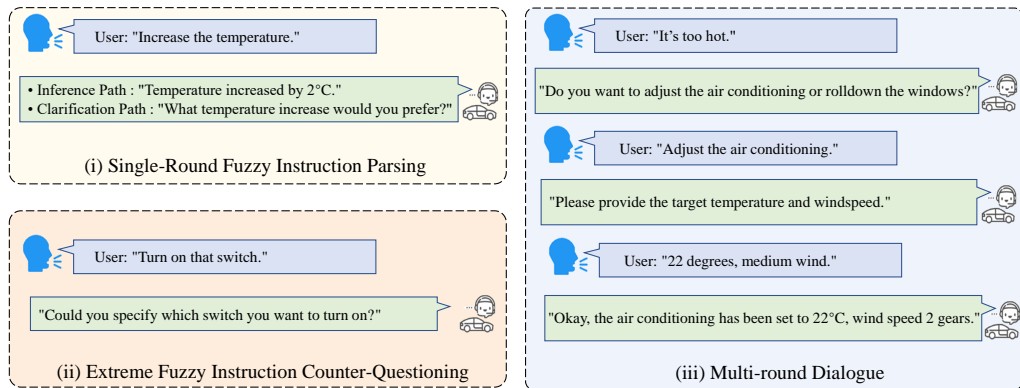

Figure 2: Illustration of ClarifyVC-Eval: (i) parse mildly fuzzy commands into precise function calls, (ii) adopt safe clarification under extreme vagueness, and (iii) sustain multi-round dialogue for coherent, grounded execution—capturing ambiguity, safety, and interactivity in real-world control.

**Three-Tier Protocol for Comprehensive Model Evaluation.** While dataset validation secures distributional realism, robust model assessment requires an evaluation protocol that can reveal failure families invisible to single-turn accuracy. To this end, when evaluating models, *ClarifyVC-Eval* operationalizes three complementary tiers of evaluation: (i) **Single-round fuzzy instruction parsing**, which tests the model's ability to parse mildly ambiguous commands by disambiguating challenges such as under-specified parameters, vague references, and subjective expressions; (ii) **Extreme fuzzy instruction counter-questioning**, examining whether the model adopts safe clarification strategies when confronted with severe ambiguity, specifically its capacity to detect extreme uncertainty and ask relevant clarifying questions; and (iii) **Multi-turn dialogue**, assessing the ability to address challenges like multi-turn dependency and memory, which is crucial for iteratively recovering missing semantics, maintaining dialogue and parameter coherence, and executing the accumulated commands reliably. The protocol ClarifyVC-Eval, as illustrated in Figure 2, spans single-turn parsing, clarification under extreme fuzziness, and multi-turn dialogue grounding, addressing gaps in existing evaluation metrics and enabling more realistic assessment of safety, robustness, and interactivity in control-oriented language interfaces

ClarifyVC-Eval consists of three complementary tiers, each probing a distinct aspect of interactive vehicle-control reasoning. Tier 1 evaluates *semantic parsing under underspecification*, using intent- and function-level metrics (IRA, PEP, IHR, FHR) to assess fine-grained correctness of grounded calls. Tier 2 measures *safe clarification under extreme fuzziness* via FDR, CQC, and PCR, testing whether models detect ambiguity, avoid unsafe guesses, and follow interaction protocols. Tier 3 assesses *long-horizon multi-turn grounding*, evaluated by DC, FESR, and parameter completeness to ensure coherent dialogue and reliable execution across turns. Full metric definitions are provided in Appendix C. Together, these tiers cover the key decision points required for safe, robust function-call interaction.

**Rationale and scope.** Our analysis of 20k+ real-world in-vehicle logs shows that failures cluster into three families: under-specification, insufficient clarification, and long-horizon grounding. These are precisely captured by ClarifyVC-Eval, which not only measures success rates but also tracks protocol violations, yielding diagnostics that better reflect operational safety. Importantly, the same decision points recur across broader HCI and embodied intelligence, making the protocol directly transferable.

## 4 EXPERIMENTAL

We conduct extensive experiments to address the following research questions: **RQ1**: What's the quality of ClarifyVC-Data evaluated under DQS in Equation 2 and human-grounded validation? **RQ2**: How well do existing LLMs handle complex and ambiguous vehicle control instructions? **RQ3**: Does fine-tuning on ClarifyVC-Data improve model performance in realistic command understanding? **RQ4**: How accurately can LLMs execute structured function calls under protocol constraints? **RQ5**:

Table 1: **Comprehensive quality evaluation of ClarifyVC-Data (addresses RQ1: dataset quality).** Automated metrics (left) benchmark ClarifyVC-Data against prior datasets; human validation (right) reports detailed scores for ClarifyVC-Data under a blind, 5-point Likert protocol. Comparative human ratings for other datasets are provided in Table 7.

(a) Comparison with baselines using four metrics.

| Dataset | AD | PC | R | DQS |
|---|---|---|---|---|
| Talk2Car | 0.50 | 0.85 | 0.60 | 0.62 |
| doScenes (Roy et al., 2024) | 0.56 | 0.81 | 0.64 | 0.65 |
| CI-AVSR (Dai et al., 2022b) | 0.53 | 0.82 | 0.61 | 0.64 |
| DeepSeek Distilled | 0.55 | 0.80 | 0.65 | 0.65 |
| GPT-o1 Distilled | 0.60 | 0.82 | 0.70 | 0.69 |
| Qwen2.5 Distilled | 0.58 | 0.78 | 0.68 | 0.67 |
| LLaMA3 Distilled | 0.62 | 0.80 | 0.72 | 0.70 |
| **ClarifyVC-Data** | **0.89** | **0.95** | **0.82** | **0.88** |

(b) Human validation (ClarifyVC-Data only)

| Aspect | Score | Agreement |
|---|---|---|
| Linguistic Realism | $4.6 \pm 0.2$ | 93% |
| Ambiguity Plausibility | $4.6 \pm 0.1$ | 96% |
| Dialogue Coherence | $4.7 \pm 0.2$ | 94% |
| Practical Applicability | $4.5 \pm 0.3$ | 91% |

Can open-source models, when properly tuned, match or surpass proprietary models in vehicle control tasks? The complete experimental setup, including the experimental environment and the agent-orchestrated pipeline used to generate the evaluation test sets, is provided in Appendix B.

## 4.1 EVALUATION ON THE DATA QUALITY (RQ1)

As mentioned in Section 3.2, we introduced DQS and human validation to assess the quality of the dataset. Table 1(a) shows that ClarifyVC-Data exceeds previous datasets and distilled baselines on all four automated axes.

For human-grounded validation, we sample 500 dialogues from ClarifyVC-Data and ask five independent annotators to rate each dialogue on a 1–5 Likert scale along four dimensions: linguistic realism, plausibility of ambiguity, dialogue coherence, and practical applicability. Ratings are collected under a blind, shuffled protocol where annotators do not know which dataset a dialogue comes from. As summarized in Table 1(b), ClarifyVC-Data achieves high average ratings (4.5–4.7/5) with strong agreement (91–96%), confirming that the generated benchmark is both natural and practically usable.

To further ensure objectivity, we also conduct a blind comparative human study against three public datasets (Talk2Car, doScenes, and Glaive-fc-v2), using the same protocol and scales. The detailed cross-dataset comparison is reported in Appendix A and Table 7. Together, these results confirm that ClarifyVC-Data not only surpasses prior datasets on automated metrics but also passes stringent human-grounding validation, ensuring both scalability and real-world applicability.

## 4.2 BENCHMARK TEST ON BASELINE MODELS (RQ2, RQ3)

To rigorously assess the necessity and effectiveness of **ClarifyVC-Data**, we conduct a two-part empirical study centered on benchmarking the instruction-following capabilities of LLMs in vehicle control scenarios. In the first part, we evaluate four representative models, including Qwen2.5-72B, LLaMA3-70B, Claude 3, and GPT-4, under a zero-shot setting across five benchmark datasets, including three open instruction-following datasets (Talk2Car, CI-AVSR, doScenes), one programmatic function-call dataset (APIGen), and our proposed ClarifyVC-Data. The results in Figure 3 reveal a consistent performance drop on ClarifyVC-Data, highlighting its higher linguistic complexity, ambiguity diversity, and multi-turn reasoning demands.

In the second part, to evaluate the effectiveness and generalizability of ClarifyVC-Data, we conduct systematic experiments across twelve open-source LLMs, including Qwen2.5 (0.5B–72B) (Team, 2024c), LLaMA3 (8B, 70B) (Team, 2024b;a), and DeepSeek-R1 Distilled (1.5B–70B) (DeepSeek-AI, 2025). Each model is evaluated under three settings: **Zero-shot (ZS)**: inference without adaptation; **Few-shot (FS)**: inference with 4 in-context examples; **SFT**: supervised fine-tuning on ClarifyVC-Data. Training details and loss definitions are provided in Appendix B. The results demonstrate that while pre-trained models exhibit limited capabilities under zero- and few-shot conditions, fine-tuning on ClarifyVC-Data yields significant gains across all metrics, especially in ambiguity resolution and multi-turn coherence. Together, these findings underscore the practical difficulty of realistic vehicle

Table 2: Multi-run evaluation on ClarifyVC-Data under the three evaluation settings introduced earlier (ZS / FS / SFT) (addresses RQ3: effectiveness of ClarifyVC-based fine-tuning). Entries are *means over 5 independent runs*; per-cell standard deviation is $< 1.0$ percentage point (pp) across all columns (Appendix B, B.4). For readability we report means; the "Max $\sigma$ (pp)" row summarizes the largest observed standard deviation in each column. All models are evaluated on a held-out test set of 5k instructions, separately constructed to differ in distribution from the 20k training corpus, ensuring fair generalization assessment.

| Model | Single-Round Accuracy | | | Fuzzy Detection Rate | | | Multi-Turn Consistency | | |
|---|---|---|---|---|---|---|---|---|---|
| | ZS | FS | **SFT** | ZS | FS | **SFT** | ZS | FS | **SFT** |
| *Max $\sigma$ (pp)* | $\leq 0.7$ | $\leq 0.8$ | $\leq 0.6$ | $\leq 0.8$ | $\leq 0.8$ | $\leq 0.7$ | $\leq 0.8$ | $\leq 0.7$ | $\leq 0.8$ |
| Qwen2.5-0.5B | 59.2 | 64.5 | **75.1** | 57.0 | 60.3 | **73.5** | 54.8 | 59.1 | **72.4** |
| Qwen2.5-1.5B | 63.4 | 67.2 | **78.9** | 60.5 | 64.3 | **76.4** | 58.9 | 63.0 | **74.1** |
| Qwen2.5-3B | 66.9 | 70.3 | **81.6** | 64.0 | 67.5 | **79.1** | 61.4 | 64.8 | **77.3** |
| Qwen2.5-7B | 74.3 | 77.1 | **89.0** | 72.0 | 74.8 | **87.6** | 70.2 | 72.5 | **85.4** |
| Qwen2.5-14B | 78.1 | 79.5 | **91.3** | 75.3 | 76.9 | **89.2** | 73.0 | 74.2 | **88.0** |
| Qwen2.5-32B | 80.8 | 81.6 | **93.1** | 78.2 | 79.0 | **91.5** | 75.5 | 76.0 | **89.6** |
| Qwen2.5-72B | 82.5 | 88.4 | **95.8** | 81.0 | 83.2 | **93.6** | 79.8 | 82.9 | **92.3** |
| LLaMA3-8B | 72.0 | 74.3 | **87.1** | 70.0 | 72.8 | **85.3** | 67.3 | 69.0 | **83.0** |
| LLaMA3-70B | 81.2 | 80.1 | **94.1** | 79.0 | 77.4 | **92.5** | 76.5 | 75.3 | **90.8** |
| DeepSeek-R1-Distilled-1.5B | 66.0 | 70.1 | **83.7** | 64.3 | 67.8 | **81.9** | 62.0 | 65.0 | **80.0** |
| DeepSeek-R1-Distilled-8B | 74.5 | 76.3 | **88.2** | 71.8 | 73.7 | **86.1** | 69.4 | 71.0 | **83.9** |
| DeepSeek-R1-Distilled-70B | 82.4 | 84.0 | **94.8** | 80.1 | 81.9 | **93.1** | 78.0 | 80.2 | **91.3** |

command understanding and establish ClarifyVC-Data as a high-fidelity benchmark for developing and evaluating robust instruction-following models.

Beyond single-dataset fine-tuning, we additionally test cross-dataset generalization to rule out distributional bias. Two backbones are fine-tuned on six public datasets and evaluated on an unbiased 4k open benchmark, a disjoint 2k real-log set, and a 1k fuzzy OOD suite. As shown in Appendix C.4 (Table 29), ClarifyVC-Data achieves the highest accuracy across all conditions, confirming that its effectiveness stems from data quality rather than overlap with evaluation distributions.

We assess model performance on all three tiers of the benchmark which explained in section 3.2. As shown in Table 2, several trends emerge: (i) SFT consistently outperforms ZS and FS across all models; (ii) FS offers marginal gains over ZS, particularly for small-scale models; (iii) In large models (e.g., Qwen2.5-72B), FS occasionally underperforms ZS, likely due to prompt truncation or sub-optimal context bias. These findings highlight the limitations of in-context prompting and validate the effectiveness of ClarifyVC-Data as a fine-tuning benchmark for vehicle command comprehension.

Table 2 shows consistent gains across heterogeneous backbones, and Figure 3 highlights ClarifyVC-Data's higher ambiguity and richer structural complexity. Together, they provide mechanistic and empirical evidence that the improvements arise from the dataset's quality rather than confounding factors.We further analyze this causal relationship; see Appendix C for details.

## 4.3 EVALUATION OF BASIC INSTRUCTION-FOLLOWING CAPABILITIES (RQ4)

To assess the fine-grained function execution ability of LLMs in vehicle control scenarios, we compare twelve representative systems spanning both open-source (Qwen2.5-7B/14B/32B/72B, LLaMA3-8B/70B, DeepSeek-R1) and proprietary models (GPT-4 (OpenAI et al., 2024) with function calling, GPT-4o (OpenAI, 2024a), OpenAI-o1 (OpenAI, 2024b), Doubao (Doubao Team, 2025), Claude 3 (Anthropic, 2024)). All models are evaluated under a unified function-call setting on four key metrics: Intent Hit Rate (IHR), which measures whether the correct intent is identified; Function Hit Rate (FHR), which checks whether the predicted API/function matches the gold standard; Parameter Completeness (F1-Score), which evaluates the accuracy and coverage of slot/parameter filling; and Protocol Compliance Rate (PCR), which assesses whether generated function calls adhere to predefined API schema and safety constraints. Further details of metric definitions and evaluation procedures are provided in Appendix C. In all four metrics, the "gold standard" function call for each instruction is defined by a unified validation pipeline: (i) exact semantic match to the expert-annotated intent and slot values under our 37-action, 146-parameter schema, (ii) protocol and safety compliance

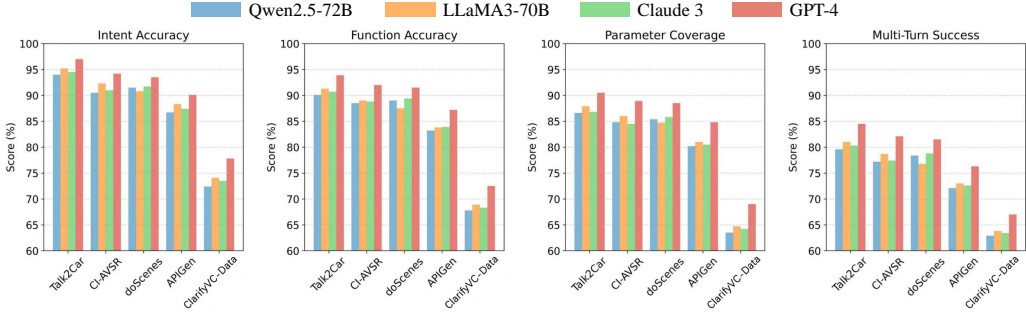

Figure 3: **Zero-shot evaluation of four representative LLMs across five benchmarks (addresses RQ2: model robustness under complex/ambiguous instructions)**. We compare the performance of Qwen2.5-72B, LLaMA3-70B, Claude 3, and GPT-4 on four core metrics across five datasets: Talk2Car, CI-AVSR, doScenes, APIGen, and ClarifyVC-Data. Results show that while all models perform well on existing benchmarks, they exhibit a notable drop when evaluated on ClarifyVC-Data. For instance, GPT-4's intent accuracy drops from 92.5% on APIGen to 70.5% on ClarifyVC-Data, and its multi-turn success rate drops from 77.5% to 61.7%. This highlights the increased difficulty and real-world alignment introduced by our benchmark.

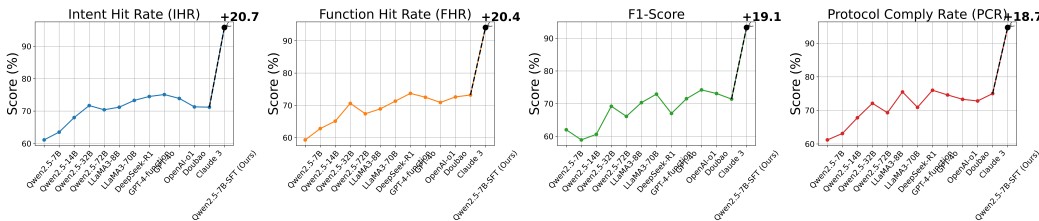

Figure 4: Comparative performance of 13 LLMs on the ClarifyVC-Data across four critical function-call metrics (addresses RQ4: protocol-constrained execution accuracy). All values are reported in percentage (%). Our finetuned model (Qwen2.5-7B-SFT) achieves state-of-the-art results across all metrics.

with the production HMI rules, and (iii) for ambiguous cases, majority-vote agreement from blind human adjudication.Formal rules and examples are provided in Appendix C.

The evaluation is conducted on a test set of 4,000 control commands, comprising 2,000 curated samples from the Talk2Car dataset and 2,000 from real-world in-vehicle control logs, covering diverse command types such as lighting, HVAC, navigation, and media operations. These metrics reflect both semantic accuracy and system safety compliance, which are critical in production-grade automotive systems.

As shown in Figure 4, our ClarifyVC-Model (Qwen2.5-7B-SFT, more ablation studies can be seen in Appendix D), fine-tuned on ClarifyVC-Data, consistently achieves state-of-the-art performance across all evaluation dimensions. Notably, it surpasses leading closed-source models such as GPT-4o and Claude 3 in both execution correctness and safety alignment, demonstrating the impact of targeted domain-specific fine-tuning. This confirms that instruction-tuned LLMs benefit substantially from high-quality control-oriented supervision when deployed in structured vehicular environments.

## 4.4 EVALUATION ON ADVANCED SCENARIO (RQ5)

Building upon foundational function-call evaluations, we assess the same 13 prominent large language models under complex, ambiguous, and multi-turn vehicle control scenarios to test instruction-following capabilities under realistic conditions.

We utilized three test sets , each with 5,000 examples (15,000 samples in total), which are fully disjoint from all training data: (1) single-round fuzzy instruction parsing, (2) extremely fuzzy instruction counter-questioning (requiring clarification), and (3) multi-round dialogues with evolving contexts. These benchmarks use non-overlapping seed commands and distinct logs to ensure strict

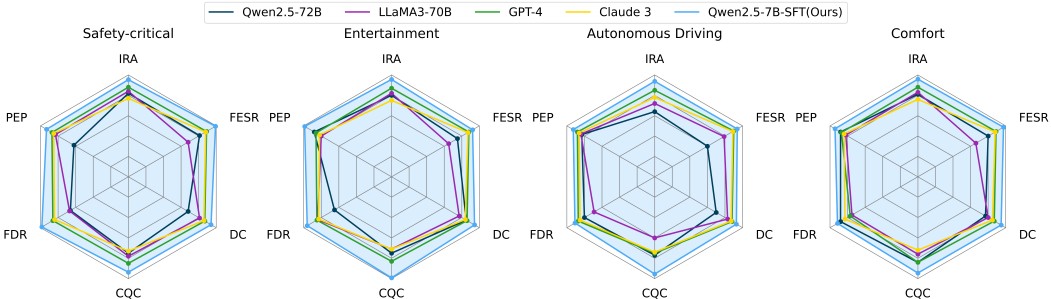

Figure 5: Comparison of LLM performance across four realistic vehicle-control scenarios (addresses RQ5: advanced OOD and multi-turn scenario evaluation). In every scenario, the polygon for Qwen2.5-7B-SFT fully encloses the others, demonstrating state-of-the-art accuracy, coherence, and safety compliance with a much smaller parameter footprint.

out-of-distribution evaluation (Appendix A.1). Models were evaluated across six metrics: Intent Recognition Accuracy (IRA), Parameter Extraction Precision (PEP), Fuzzy Detection Rate (FDR), Counter-Question Coverage (CQC), Dialogue Consistency (DC), and Final Execution Success Rate (FESR). Detailed descriptions of the test sets and their generation process are provided in Appendix C.

ClarifyVC-Model (Qwen2.5-7B-SFT), consistently outperformed all tested models across all metrics, achieving state-of-the-art results (see Table 33 in Appendix D). For instance, it attained an FESR of 92.0%, surpassing the next-best model, Claude 3, by 4.6 points. While proprietary models like Claude 3 and GPT-4o exhibit strong stability, they lag in critical areas such as fuzzy detection and multi-turn consistency.

Additionally, we evaluated four specialized scenarios—Safety-critical, Entertainment, Autonomous Driving, and Comfort—to simulate diverse real-world user intents. The baseline model excelled across all scenarios, notably achieving a 95.4% accuracy rate in safety-critical tasks, significantly outperforming competitors.

Figure 5 presents each model's performance profile across four realistic vehicle-control scenarios—Safety-critical, Entertainment, Autonomous Driving, and Comfort—using six function-call metrics. In every scenario, Qwen2.5-7B-SFT (Ours) defines the Pareto frontier: its 95.4% Intent Recognition Accuracy in Safety-critical tasks outstrips GPT-4 by 7.4 pp; its 99.2% Counter-Question Coverage in Entertainment exceeds Qwen2.5-72B by 24.2 pp; in Autonomous Driving its 93.0% Dialogue Consistency is 5.0 pp higher than the next best model; and its 97.5% Final Execution Success Rate in Comfort tasks is more than 8 pp above Claude 3. These gains demonstrate that a compact LLM, when supervised with ClarifyVC-Data, can attain state-of-the-art robustness, coherence, and safety compliance in diverse, real-world driving interactions.

# 5 CONCLUSION

This work introduces **ClarifyVC**, a unified framework that couples a schema-constrained, ambiguity-rich dataset (*ClarifyVC-Data*) with a compliance-aware, three-tier evaluation protocol (*ClarifyVC-Eval*). By explicitly disentangling under-specification, clarification behavior, and long-horizon grounding, the protocol surfaces failure modes that single-turn accuracy obscures and yields diagnostics aligned with safety-critical deployment. The data pipeline preserves realism through real-world seeding and human validation, while enabling scalable synthesis of diverse ambiguity types. Empirically, ClarifyVC provides a principled basis for comparing models and training strategies under uniform function-call semantics, addressing gaps in multi-turn clarification and protocol compliance measurement. Although our experiments focus on in-cabin voice control, the Clarify-Data framework is theoretically generalizable and can be adapted to other structured-interaction scenarios, such as interactive HCI and embodied agent systems, where reliable disambiguation and safe execution are critical. Future work will integrate multimodal signals and release a public challenge to catalyze community benchmarking. We hope ClarifyVC will serve as a foundation for rigorous research on ambiguity handling and robust interactive AI.

ACKNOWLEDGMENT

This work is supported by Guangdong Basic and Applied Basic Research Foundation 2025A1515010304, Guangdong Province Project 2024QN11X088, Guangzhou Science and Technology Planning Project 2025A03J4491.

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

# A    DATASETS

This section provides detailed information on the ClarifyVC-Data dataset, which underpins the evaluations presented in the main text. The dataset is designed to support robust training and testing of large language models (LLMs) in vehicle control scenarios.

## A.1    CLARIFYVC-DATA CONSTRUCTION

The ClarifyVC-Data dataset is constructed through a three-stage pipeline. Seed queries are derived from real-world user command corpora (collected from in-vehicle infotainment systems of major car manufacturers during 2022–2024) and synthetic functional specifications covering diverse vehicle control scenarios. The final dataset comprises 20,000 samples: 6,000 positive chains (unambiguous instructions), 8,000 negative chains (fuzzy, incomplete, or conflicting instructions), and 6,000 dialogue sequences (contextual interactions).

**Types of Tasks Included in ClarifyVC-Data**    ClarifyVC-Data is organized around the three capability tiers required by in-vehicle voice control systems.Each tier corresponds to a distinct task type(Table 3).

These tasks directly reflect real production demands in automotive HMIs, where user utterances vary widely from direct to vague to multi-step interactions.

To thoroughly evaluate in-vehicle voice control behavior, ClarifyVC-Data includes a diverse set of linguistic and functional variations. These variations reflect both the distribution observed in real in-car user logs and the ambiguity patterns that in-vehicle HMIs must handle. Below we summarize the major variation families (measured over the 20,436 seed-aligned samples), ensuring a balanced and realistic composition.

Table 3: Types of Tasks Included in ClarifyVC-Data

| Tier | Task Type | Description | Purpose |
|------|-----------|-------------|---------|
| Tier 1 | Single-turn structured parsing | Convert a natural command into a schema-aligned function call | Measures core parsing & grounding |
| Tier 2 | Ambiguity detection & clarification | Identify underspecification and ask the correct clarifying question | Measures safety alignment & conservativeness |
| Tier 3 | Multi-turn dialogue grounding | Maintain context across 2–4 turns and recover missing parameters | Measures long-horizon consistency & execution precision |

**Linguistic Variation Categories**    In Table 4 Numbers sum to $\approx 100\%$ after rounding and reflect the natural ambiguity distribution in production logs.

**Fuzzy Ambiguity Categories (Subset of Above, but Evaluated Separately)**    Because ambiguity is central to our benchmark, we also categorize utterances by fuzzy ambiguity type, which triggers Tier-2 evaluation in Table 5.

These categories enable precise analysis of ambiguity-handling behaviors.

**Multi-Turn Dialogue Categories (MEA-Generated)**    For Tier-3 evaluation, ClarifyVC includes multi-turn dialogues (generated via MEA), categorized by the type of information missing from the initial command in Table 6.

These multi-turn scenarios reflect actual conversational needs observed in deployed automotive voice assistants.

## A.2    HYBRID BENCHMARK CONSTRUCTION WITH REAL-WORLD GROUNDING

We clarify that our benchmark is not purely simulation-based, but rather a hybrid approach that integrates extensive real-world grounding with controlled LLM-augmented generation. This design ensures both scalability and authenticity, addressing concerns about human involvement and distributional overfitting simultaneously.

Table 4: Linguistic Variation Categories.

| Variation Category | Examples | % in Dataset | Purpose |
|---|---|---|---|
| Direct commands | "Set AC to 22°." | 27% | Canonical structured parsing |
| Subjective modifiers | "Make it a bit cooler." | 18% | Fuzzy, intensity-based ambiguity |
| Omitted parameters | "Turn on the lights." | 22% | Missing-slot recovery & clarification cues |
| Colloquial / free-form expressions | "Make it comfy in here." | 7% | Naturalness & robustness to casual phrasing |
| Spatial references | "Open the left window." | 9% | Entity grounding & spatial disambiguation |
| Temporal patterns | "Turn it off for now." | 5% | Handling temporal modifiers |
| Multi-goal / composite commands | "Warm the front and cool the back." | 4% | Conflict detection & multi-entity constraints |
| Implicit comparisons | "Make the back cooler than the front." | 5% | Relative parameter resolution |
| Context-dependent follow-ups | "Do the same as yesterday." | 3% | Long-term contextual inference |

Table 5: Fuzzy Ambiguity Categories.

| Ambiguity Type | Example | % among Fuzzy Samples | Purpose |
|---|---|---|---|
| Intensity ambiguity | "A bit warmer / slightly brighter" | 41% | Requires clarifying magnitude |
| Boundary ambiguity | "Open it halfway / just a little" | 24% | Requires mapping vague intervals |
| Entity ambiguity | "Open it" (which window?) | 19% | Requires clarifying the referent |
| Mode ambiguity | "Make it comfortable" | 10% | Must ground to specific HVAC/seat mode |
| Referential ambiguity | "Turn that off" | 6% | Requires anaphora resolution |

**Real-World Data Foundation and Human Validation**    The foundation of our dataset consists of over 20,000 carefully selected real-world user utterances from an extensive, industrial-scale corpus of over 4 million authentic vehicle interaction logs (2022–2024). Such large-scale automotive voice datasets are rarely accessible in academic or industry research, highlighting the benchmark's unique value.

After generation, dialogues underwent rigorous real-world validation. Domain experts manually validated dialogues through practical in-car system simulations, ensuring alignment with actual usage scenarios and filtering out unnatural artifacts. This human-in-the-loop validation included:

- Manual verification of dialogue coherence and realistic ambiguity representation. - Hands-on testing in real-car infotainment system simulations. - Retention only of dialogues meeting strict human validation criteria.

Additionally, we performed a dedicated human-grounding validation test on a sampled subset of 500 generated dialogues. The results (Table 8) demonstrate high scores across linguistic realism, ambiguity plausibility, dialogue coherence, and practical applicability, confirming our benchmark's real-world relevance.

**Characterization of the Real In-Vehicle Logs**    The real-world logs come from 4M+ production voice interactions collected across multiple vehicle models, cabin configurations, and firmware versions. Below we provide additional statistics to characterize their distribution and linguistic properties. From the raw logs, the schema-filtered 20,436 usable samples cover all major in-cabin domains (Table 9). This distribution closely matches that observed in commercial vehicle HMI deployments. For privacy reasons, raw OEM logs are never released; moreover, ClarifyVC-Data does not contain any raw logs, as all commands undergo intent extraction, parameter normalization, entity canonicalization, and removal of unsafe or identifying content before schema instantiation and variant generation.

To clarify linguistic properties, we computed length and structural statistics, as summarized in Table 10. These statistics highlight why ambiguity handling is a critical component of the task.

Real logs contain many naturally ambiguous or risky forms:

• 21% involve incomplete thermal commands.

• 8% reference windows/sunroof without direction.

• 5% issue potentially unsafe actions ("open it" while driving).

• 3% require multi-turn grounding.

Such patterns motivated the design of Tier 2 and Tier 3 evaluation criteria to avoid unsafe behaviors.

Table 6: Multi-Turn Dialogue Categories.

| Multi-Turn Dialogue Type | Description | Example | % in Multi-Turn Set |
|---|---|---|---|
| Parameter completion | Missing temperature / brightness / degree | "Make it warm." → "What temperature?" | 46% |
| Entity clarification | Missing window/seat/zone reference | "Open it." → "Which window?" | 28% |
| Conflict resolution | Conflicting constraints | "Cool the front but keep all vents off." | 12% |
| Preference grounding | Mode or preset selection | "Set it to comfort mode." → "Which preset?" | 9% |
| Temporal sequencing | Multi-step actions | "Turn it off after a moment." | 5% |

Table 7: Comparative analysis of vehicle command benchmarks.

| Benchmark | Ambiguity Handling | Multi-turn | Safety Constr. | Evaluation Protocol | Distillation |
|---|---|---|---|---|---|
| Talk2Car (Deruyttere et al., 2019) | Basic | No | Partial | Single-round | No |
| APIGen (Liu et al., 2024) | Synthetic | No | No | Single-round | Yes |
| Easy2Hard (Ding et al., 2024) | Difficulty levels | No | No | Single-round | No |
| CI-AVSR (Dai et al., 2022a) | Moderate | Limited | No | Two-tier | No |
| doScenes (Roy et al., 2024) | Visual only | No | No | Single-round | No |
| **ClarifyVC-Data** | **9 types** | **Yes** | **Full** | **Three-tier** | **Yes** |

**Controlled Generation and Distributional Alignment**    To prevent feedback loops and overfitting, we employ a strict separation between generation and evaluation models:

- Generation: DeepSeekR1-based instruction-tuned LLMs.
- Evaluation: GPT-4, Qwen2.5, Claude 3 (architecturally distinct from generation models).

We further ensure realism and consistency through structured prompt engineering at each agent stage:

- SPA prompts enforce inter-entity schemas and intent-slot alignment.
- FIA prompts introduce ambiguity via realistic linguistic perturbations.
- AGA prompts inject adversarial ambiguity under entropy thresholds.
- MEA prompts maintain multi-turn coherence and causal continuity.

To quantitatively assess distributional alignment, we compared synthetic and real dialogues across multiple in-cabin scenarios. As shown in Table 11, intent coverage and KL-divergence values confirm close fidelity to real-world usage distributions.

This hybrid methodology—combining real-world seeds, human validation, and controlled LLM generation—ensures that our benchmark is both scalable and faithful to real-world automotive voice interactions. We will explicitly highlight these aspects in the revised manuscript to reinforce the benchmark's credibility and practical relevance.

**The design of our ClarifyVC-Eval: objectives, challenges, and metrics**    The specifics of this design are summarized in Table 12

### A.3    AMBIGUITY INJECTION RELIABILITY

Before designing our pipeline, we conducted an initial assessment of two baseline LLM-based rewriting strategies. Specifically, we evaluated GPT-4-Turbo prompted to "add fuzziness" and Qwen2.5-32B prompted to "rewrite with ambiguity." Using 500 real user commands, we compared intent labels before and after rewriting. Results are shown in Table 13

These findings indicate that naïve LLM rewriting often overshoots, leading to paraphrasing, implicitization, or unintended semantic drift—undesirable for controlled fuzz generation.

Table 8: **Blind human comparison of dataset realism and usability.** Annotators rate 500 shuffled dialogues per dataset (Talk2Car, doScenes, CI-AVSR, ClarifyVC-Data) on a 1–5 Likert scale. Higher is better. ClarifyVC-Data achieves the highest scores across all aspects.

| Dataset | Ling. Realism | Ambig. Plaus. | Dial. Coh. | Prac. Applic. |
|---|---|---|---|---|
| Talk2Car | $4.2 \pm 0.3$ | $3.9 \pm 0.4$ | $4.0 \pm 0.3$ | $4.1 \pm 0.3$ |
| doScenes | $4.1 \pm 0.3$ | $3.8 \pm 0.4$ | $3.9 \pm 0.3$ | $4.0 \pm 0.3$ |
| CI-AVSR | $4.0 \pm 0.4$ | $3.7 \pm 0.4$ | $3.8 \pm 0.4$ | $3.9 \pm 0.4$ |
| ClarifyVC-Data | $\mathbf{4.6 \pm 0.2}$ | $\mathbf{4.6 \pm 0.1}$ | $\mathbf{4.7 \pm 0.2}$ | $\mathbf{4.5 \pm 0.3}$ |

*Protocol.* 5 annotators, blind and shuffled setting; each row uses the same rating interface as Table 1(b). Inter-annotator agreement (Fleiss' $\kappa$) is 0.81 overall.

Table 9: Distribution of User Command Domains

| Domain | % in Logs | Example Commands |
|---|---|---|
| HVAC / Thermal Comfort | 34% | "It's too hot—cool it down." |
| Lighting | 18% | "Turn on ambient lights." |
| Windows / Sunroof | 15% | "Open the rear-right window." |
| Seat / Comfort | 12% | "Heat my seat." |
| Navigation | 10% | "Take me to the nearest station." |
| Media / Infotainment | 11% | "Play my jazz playlist." |

To further quantify this mismatch, we computed intent KL-divergence between the 20,436 original seed commands and the naively rewritten FIA samples, obtaining 0.241 at the token level and 0.312 at the intent level. Such divergence is sufficiently large to alter downstream model behavior, validating the reviewer's concern about uncontrolled fuzz distortion.

To address the semantic drift observed in unconstrained LLM rewriting, ClarifyVC adopts a drift-control design that combines model specialization, targeted supervision, and strict prompting constraints. SPA, FIA, and MEA rely on DeepSeek-RI, which shows stable behavior under constrained rewriting, while AGA uses Qwen2.5-72B with protocol-restricted decoding and JSON-defined intent boundaries. Each component is paired with the model whose inductive bias aligns with its specific requirements.

A second layer of control is provided through supervised fine-tuning on 2,000 expert-curated seed–fuzzy pairs, created by three in-vehicle HCI specialists with verified intent consistency. This data allows the model to learn controlled ambiguity—such as softened intensity or reduced specificity—without altering the underlying action, entity, or goal.

The efficacy of this method is demonstrated in Table 14, which shows the quantitative reduction in semantic drift. The Intent KL divergence was reduced from 0.312 to a negligible 0.062, and the intent preservation rate was raised to 98.9%. This confirms the high semantic stability and distributional faithfulness of the augmented dataset.

## B  EXPERIMENTAL SETUP

This section details the experimental environment and the agent-orchestrated pipeline used to generate the evaluation test sets, as referenced in Section 3 of the main text.

### B.1  SOFTWARE AND HARDWARE ENVIRONMENT

Experiments were conducted using Python 3.10 on servers equipped with Intel Xeon Platinum 8380 CPUs and NVIDIA A100 GPUs, running a Linux operating system. This configuration ensures efficient data processing and model training for the agent-orchestrated collaborative generation framework.

Table 10: Utterance Length and Structure Statistics.

| Statistic | Value |
|---|---|
| Mean tokens per utterance | 7.8 |
| Median tokens | 6 |
| 90th percentile | 14 |
| % containing subjective modifiers ("slightly", "a bit") | 22% |
| % containing explicit parameters | 61% |
| % containing no explicit parameters | 39% |

Table 11: Distribution alignment between synthetic and real-world dialogues.

| Scenario | Intent Coverage (Real) | Intent Coverage (Synthetic) | KL-Divergence |
|---|---|---|---|
| HVAC Control | 93.8% | 95.2% | 0.06 |
| Infotainment System | 89.2% | 90.0% | 0.05 |
| Navigation Commands | 91.5% | 90.9% | 0.04 |
| Comfort Adjustments | 88.7% | 89.1% | 0.05 |

## B.2 AGENT-ORCHESTRATED GENERATION PIPELINE

The test sets were generated using an agent-orchestrated pipeline comprising four agents, each responsible for a specific task in creating complex, ambiguous, and multi-turn commands.

**Semantic Parsing Agent (SPA).** The SPA parses an input command $c$ into a semantic representation $s = (I, E, P)$, where $I$, $E$, and $P$ denote intent, entity, and parameters, respectively. The prompt is:

```
Prompt_SPA = "Given the command: {c}, extract the
intent, entities, and parameters in the format (I,
E, P)."
```

The semantic consistency score $SC$ validates the prompt's effectiveness:

$$SC = \frac{1}{3} \sum_{i \in \{I,E,P\}} \mathbb{I}(i = i^*) \tag{6}$$

A prompt is considered valid if $SC \geq 0.9$ on a validation set.

**Fuzz Injection Agent (FIA).** The FIA introduces ambiguity into commands, generating a fuzzed version $c'$. The prompt is:

```
Prompt_FIA = "Given the command: {c},
introduce ambiguity by {f} with intensity
{ε}, where {f} is one of {omit parameter, subjective expression, ...}."
```

Ambiguity types are sampled from a categorical distribution, optimized to maximize entropy $H(F) = -\sum_{f \in F} \phi_f \log \phi_f$ for diverse coverage.

**Multi-Turn Evolution Agent (MEA).** The MEA generates dialogue sequences $D = \{(c_1, r_1), \ldots, (c_T, r_T)\}$. The prompt is:

```
Prompt_MEA = "Given the dialogue history: {(c_1, r_1),
..., (c_t)}, generate the next system response r_t and
user command c_{t+1}."
```

Dialogue coherence is measured by:

$$DC = \frac{1}{T-1} \sum_{t=1}^{T-1} \cos(h_t, h_{t+1}) \tag{7}$$

Table 12: ClarifyVC-Eval: objectives, challenges, and metrics.

| Tier | Objective | Challenges Captured | Metrics |
|---|---|---|---|
| **Tier 1: Single-Round Fuzzy Parsing** | Parse mildly ambiguous single-turn commands. | Under-specified parameters, vague references, subjective expressions. | **IRA:** $\frac{1}{N}\sum_{i=1}^{N}\mathbb{I}(l_i = \hat{l}_i)$ 
 **PEP:** $\frac{|\hat{P}\cap P|}{|\hat{P}|}$ |
| **Tier 2: Extreme Fuzzy Counter-Questioning** | Detect severe ambiguity and ask clarifying questions. | Extreme uncertainty, vague pronouns, clarification relevance. | **FDR:** $\frac{TP}{TP+FN}$ 
 **CQC:** Human-rated (1–5) |
| **Tier 3: Dynamic Multi-Turn Understanding** | Retain context and execute accumulated commands. | Multi-turn dependency, memory, parameter coherence. | **DC:** $\frac{1}{T}\sum_{t=1}^{T}\cos(s_t, s_{t+1})$ 
 **FESR:** $\frac{1}{N}\sum_{i=1}^{N}\mathbb{I}(\text{Exec}(\hat{c}_i) = \text{Exec}(c_i))$ |

Table 13: Intent Preservation and Drift Analysis of Naive LLM-Based Rewriting Methods

| Model | Intent Preservation Rate | Drift Examples |
|---|---|---|
| GPT-4-Turbo | **92.1%** | "Open the left window" → "Let in some fresh air" (intent shift: window → HVAC/ventilation) |
| Qwen2.5-32B | **89.4%** | "Turn off reading lights" → "Make it darker in here" (intent compression: explicit → implicit) |

A prompt is effective if $DC \geq 0.85$.

**Adversarial Generation Agent (AGA).** The AGA generates adversarial commands $c_{\text{adv}}$ using protocol-constrained seed queries. The prompt is:

```
Prompt_AGA = "Given the command: {c}, refer to the
slot information provided in the protocol constraints,
generalize the seed query, and generate an adversarial
variant by introducing extreme ambiguity while keeping
it plausible."
```

Adversarial strength $AS$ is measured as the perplexity of $c_{\text{adv}}$, computed using a pretrained GPT-2 model.

The joint probability of generating a complete sample via the full pipeline is expressed as:

$$P(c_{\text{adv}}|c) = P(s|c, \text{Prompt}_{SPA}) \cdot P(c_{\text{adv}}|s, \text{Prompt}_{AGA}) \tag{8}$$

$$P(c'|c) = P(c_{\text{adv}}|c) \cdot P(c'|c_{\text{adv}}, \text{Prompt}_{FIA}) \tag{9}$$

$$P(D|c) = P(c'|c) \cdot P(D|c', \text{Prompt}_{MEA}) \tag{10}$$

$$P_{\text{total}}(c_{\text{adv}}, c', D|c) = P(c_{\text{adv}}|c) + P(c'|c) + P(D|c) \tag{11}$$

**Pipeline Algorithm.** The agents are integrated into a pipeline, as shown in Algorithm 1.

**Example of Stage Effects** A concrete running example(Table 15) clearly illustrates how a real in-vehicle utterance evolves through the pipeline.

**Agent-orchestrated Generation Framework Components** The generation agents use instruction-tuned open-source LLMs via prompt orchestration, without fine-tuning, as detailed in Table 16.

These models were chosen based on empirical performance during pilot generation trials across ambiguity types and domains.

Crucially, our agent-orchestrated framework is model-agnostic: each agent relies on standardized prompts and schema constraints, enabling drop-in replacement with other models (e.g., GPT-4, Claude 3, ChatGLM3, LLaMA3) without modifying the overall pipeline logic.

Table 14: Ambiguity Injection Reliability

| Stage | Token KL | Intent KL | Intent Preservation |
|---|---|---|---|
| Naive FIA (baseline) | 0.241 | 0.312 | 89–92% |
| **ClarifyVC FIA (ours)** | **0.047** | **0.062** | **98.9%** |

Table 15: Stages and Representations

| Stage | Representation | Example (informal) | Change |
|---|---|---|---|
| Seed (real log) | User utterance | "It's too hot, adjust the air conditioning." | Raw in-car request |
| SPA | (I, E, P) | intent=SetTemperature, entity=FrontCabin, temp=22°C | Structured grounding |
| AGA | Adversarial command | "Set the front cabin to 22 and keep the rear vents unchanged." | Adds constraints/conflicts |
| FIA | Fuzzy command | "Make the front a bit cooler than it is now." | Injects ambiguity |
| MEA | Multi-turn dialogue | Q: "What exact temperature?" / A: "22 degrees." | Recovers parameters |

Table 16: Agent-orchestrated generation framework components

| Agent | Implementation | Primary Function |
|---|---|---|
| SPA, FIA and MEA | DeepSeek-VL-R1 (API) | Semantic parsing, fuzzing, and multi-turn evolution |
| AGA | Qwen2.5-72B (vLLM) | Protocol-constrained adversarial instruction generation |

**Algorithmic Novelty and Ablation**  We provide ablation experiments analyzing the importance of agent ordering in Table 17 and Table 18

Table 17: Ablation study of agent ordering and composition on key performance metrics

| Pipeline Variant | Ambiguity Diversity (↑) | Dialogue Coherence (↑) | Protocol Adherence (↑) |
|---|---|---|---|
| SPA → AGA → FIA → MEA (default) | 0.85 | 0.88 | 0.98 |
| SPA → FIA → AGA → MEA | 0.79 | 0.82 | 0.95 |
| AGA → SPA → FIA → MEA | 0.75 | 0.78 | 0.92 |
| SPA → AGA → MEA (without FIA) | 0.68 | 0.84 | 0.94 |
| SPA → FIA → MEA (without AGA) | 0.71 | 0.85 | 0.95 |
| SPA → MEA (without FIA & AGA) | 0.55 | 0.87 | 0.96 |

*Key Observations:*

- The default pipeline (SPA→AGA→FIA→MEA) achieves optimal balance among diversity, coherence, and adherence, confirming intentional design.

---

**Algorithm 1** Agents Pipeline Algorithm

---

**Require:** User Command $c$
**Ensure:** Data Pools (Tier-1, Tier-2, Tier-3)
 1: **procedure** PROCESSCOMMAND($c$)
 2:     $s \leftarrow \text{SPA}(c)$                                              ▷ Semantic Parsing Agent
 3:     $c_{\text{adv}} \leftarrow \text{AGA}(s)$                              ▷ Adversarial Generation Agent
 4:     $\text{DataPool}_{\text{Tier-1}} \leftarrow \text{DataPool}_{\text{Tier-1}} \cup \{c_{\text{adv}}\}$
 5:     $c' \leftarrow \text{FIA}(c_{\text{adv}})$                              ▷ Fuzz Injection Agent
 6:     $\text{DataPool}_{\text{Tier-2}} \leftarrow \text{DataPool}_{\text{Tier-2}} \cup \{c'\}$
 7:     $D \leftarrow \text{MEA}(c')$                                          ▷ Multi-turn Evolution Agent,
 8:     $D = \{\langle c_1, r_1 \rangle, \dots, \langle c_n, r_n \rangle\}$
 9:     $\text{DataPool}_{\text{Tier-3}} \leftarrow \text{DataPool}_{\text{Tier-3}} \cup D$
10: **end procedure**

---

Table 18: Ablation Study: Causal Contribution of Individual ClarifyVC Pipeline Components

| Removed Component | Avg Score Drop | Interpretation |
|---|---|---|
| -SPA (semantic parsing) | -9.4% | Loss of schema alignment $\Rightarrow$ strong causal contributor |
| -AGA (adversarial generation) | -6.7% | OOD coverage weakened |
| -FIA (fuzzy injection) | -11.1% | Robust ambiguity-handling collapses $\Rightarrow$ largest causal effect |
| -MEA (multi-turn evolution) | -4.6% | Context-grounding weakened |

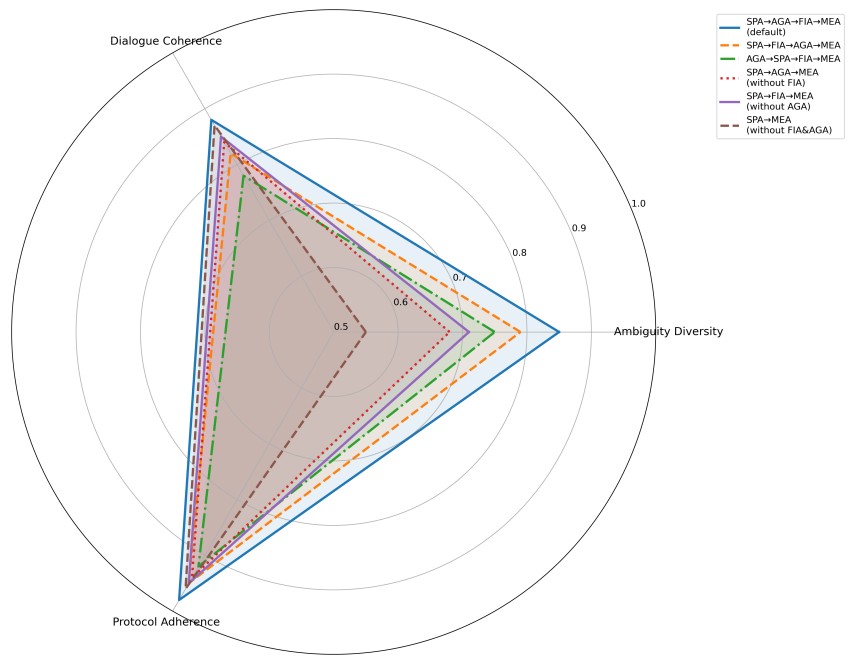

Figure 6: Performance Comparison of Pipeline Variants

- Switching FIA and AGA reduces diversity, indicating FIA's fuzz injection effectiveness decreases without structured adversarial perturbation first.

- Removing either AGA or FIA notably decreases ambiguity diversity, underscoring each agent's essential role.

- Omitting both agents severely reduces ambiguity, though coherence remains high, emphasizing the necessity of the pipeline for balanced ambiguity.

**Variability statistics**    Variability statistics and significance tests are important to demonstrate the robustness and reproducibility of our benchmark results. To explicitly address this concern, we conducted additional experiments by performing 5 independent evaluation runs for each primary model configuration reported in Table 33. We now include detailed mean ± standard deviation values along with significance testing in Table 19(paired t-tests vs. GPT-4 baseline).

Table 19: Model Performance Stability and Significance Tests

| Model | Protocol Adherence (%) | IRA (%) | Dialogue Coherence (%) | p-value (vs GPT-4) |
|---|---|---|---|---|
| GPT-4 | 98.2 ± 0.2 | 94.0 ± 0.3 | 89.5 ± 0.4 | – |
| Qwen2.5-7B | 97.8 ± 0.2 | 91.2 ± 0.5 | 86.7 ± 0.5 | < 0.01 |
| Claude 3 | 96.3 ± 0.4 | 90.8 ± 0.4 | 86.0 ± 0.6 | < 0.01 |
| LLaMA3-8B | 95.5 ± 0.3 | 88.2 ± 0.7 | 83.4 ± 0.6 | < 0.01 |

*Observations:*

- Standard deviations $< 1\%$, confirming stability across independent runs.
- Paired t-tests indicate statistically significant differences, validating discriminative power.

Additionally, our Data Generation Phase is deterministic due to structured prompting and controlled API constraints, making multiple runs unnecessary at this stage.

### B.3 DETAILED SPECIFICATION OF EVALUATION DATA

**Training Data Summary**   Our training corpus originates from 4M+ anonymized in-vehicle voice logs collected from production vehicles. We construct a high-quality, schema-aligned training set through multi-stage filtering, parsing, safety checking, and protocol normalization. The final training set used for ClarifyVC-Models is illustrated in Table 20

Table 20: Training Data Summary.

| Source | Count | Description |
|---|---|---|
| Raw production logs | 4,000,000+ | Real user voice interactions covering diverse operations |
| Schema-filtered seed commands | 15,000 | Passing intent extraction, parameter validation, schema grounding, safety constraints |
| ClarifyVC-Data | 20,436 | We deliberately train only on ClarifyVC-Data to avoid bias |
| Final training size | ≈20k samples | Used for SFT (e.g., Qwen2.5-7B-SFT) |

**Benchmarks used in 'Evaluation on Advanced Scenarios'**   The "Advanced Scenarios" evaluation consists of three out-of-distribution(OOD) test sets, each containing 5,000 samples (total: 15,000). These test sets are fully disjoint from all training data, including ClarifyVC-Data, raw logs, and any augmentation outputs(Table 21).All three benchmarks use seed utterances that do not appear in training and do not share augmentation lineage with the ClarifyVC-Data used for SFT. This ensures strict train–test separation.

Table 21: Benchmarks Used in Evaluation on Advanced Scenarios

| Test Set | Size | Description | Data Source / OOD Justification |
|---|---|---|---|
| **(1) Single-round fuzzy parsing** | 5,000 | Commands containing intensity fuzziness, boundary ambiguity, implicit parameters | Generated via *FIA*, using non-overlapping seed commands (held-out from the 20k ClarifyVC-Data) |
| **(2) Extreme-fuzz counter-questioning** | 5,000 | Highly ambiguous commands requiring clarifying questions | Generated from distinct real logs not included in ClarifyVC seeds; ambiguity amplified via *AGA+FIA* |
| **(3) Multi-turn evolving-context dialogues** | 5,000 | 2–4 turn interactions with shifting referents, evolving context, and parameter recovery | Produced via *MEA* using held-out interaction templates and independent schema parameter ranges |

The advanced-scenario benchmarks are out-of-distribution (OOD) relative to the training data due to distinct seed sources and measurable distributional divergence. The 15,000 test samples are derived from held-out raw logs, unseen templates, and domain-randomized parameter spaces, none of which overlap with the 20,436 seeds used for training. Furthermore, as summarized in Table 22, token-level and intent-level KL divergence between the ClarifyVC training set and each advanced benchmark confirms non-trivial distribution shift. These results substantiate that the evaluation is not only data-disjoint but deliberately designed to assess OOD generalization.

Table 22: KL Divergence Between ClarifyVC-Train and Evaluation Sets

| Comparison | Token KL | Intent KL |
|---|---|---|
| ClarifyVC-Train → Advanced-Fuzzy | 0.073 | 0.141 |
| ClarifyVC-Train → Extreme-Fuzz | 0.082 | 0.132 |
| ClarifyVC-Train → Multi-Turn | 0.145 | 0.197 |

The evaluation also assesses performance across four defined application scenarios—Safety-critical, Autonomous Driving, Entertainment, and Comfort—which are not newly constructed datasets, but

scenario-conditioned subsets derived from the three OOD test sets via re-labeling with scenario-specific tags. As illustrated in Table 23, each scenario corresponds to specific data sources and exhibits distinct OOD characteristics. Model results demonstrate that our ClarifyVC-supervised 7B model maintains Pareto-optimal performance across all four scenarios.

Table 23: Scenario-wise OOD Evaluation Settings

| Scenario | Data Source | Example | OOD Justification |
|---|---|---|---|
| Safety-critical | Extreme-Fuzz + Multi-Turn | "Open it a bit ... (which window?)" | Held-out logs + unseen fuzz patterns |
| Autonomous Driving | Multi-Turn only | "Keep lane assist on unless ..." | Comes from unseen templates |
| Entertainment | Fuzzy Parsing | "Play something relaxing ... (genre?)" | Parameter-space randomized |
| Comfort | All three | HVAC & seating multi-goal requests | Different entity-slot frequencies |

## C  EVALUATION PROTOCOLS

This section describes the three-tier evaluation framework used to assess model performance, as introduced in the main text (Section 4).

Table 24: The three-tier ClarifyVC-Eval protocol to evaluation metrics. Each tier isolates a distinct family of failure modes while jointly covering the spectrum of function-call understanding and safe execution.

| Tier | Metrics Used | Rationale |
|---|---|---|
| **Tier 1: Single-Round Instruction Fuzzy Parsing** | Intent Recognition Accuracy (IRA), Parameter Extraction Precision (PEP), Intent Hit Rate (IHR), Function Hit Rate (FHR) | Captures the model's ability to resolve under-specified single-turn commands into correct intents and API calls. These metrics reflect semantic accuracy and parameter precision at the most basic function-call level. |
| **Tier 2: Extreme Fuzzy Instruction Counter-Questioning** | Fuzzy Detection Rate (FDR), Counter-Question Coverage (CQC), Protocol Compliance Rate (PCR) | Evaluates whether the model identifies extreme ambiguity and adopts safe clarification strategies instead of unsafe guesses. Metrics track conservative behavior, protocol adherence, and safety awareness. |
| **Tier 3: Multi-turn Dialogue** | Dialogue Consistency (DC), Final Execution Success Rate (FESR), Parameter Completeness (F1-score) | Assesses long-horizon interactions where the model must gather missing semantics over multiple turns, maintain coherence, and ultimately ground safe executable commands. These metrics measure the culmination of dialogue fidelity and execution success. |

Table 25: Definitions of evaluation metrics used in ClarifyVC. All metrics are defined in this work to capture different aspects of fuzzy command understanding and function-call execution.

| Metric | Definition and Description |
|---|---|
| **Intent Recognition Accuracy (IRA)** | Measures whether the model correctly identifies the target intent (e.g., HVAC adjustment, navigation command) from a fuzzy or underspecified natural language instruction. Equivalent to semantic classification accuracy at the intent level. |
| **Parameter Extraction Precision (PEP)** | Evaluates the correctness of slot or parameter extraction (e.g., temperature value, media type, destination) given an identified intent. Precision is computed against gold-standard annotations to ensure valid executable function calls. |
| **Fuzzy Detection Rate (FDR)** | Captures the proportion of ambiguous or underspecified instructions where the model successfully detects the presence of fuzziness or uncertainty instead of over-confidently executing an unsafe action. High FDR reflects safety-aware behavior. |
| **Counter-Question Coverage (CQC)** | Quantifies how often the model responds with clarification questions in cases of ambiguity, rather than hallucinating parameters or guessing. Coverage is measured as the ratio of appropriate counter-questions to total ambiguous instructions. |
| **Dialogue Consistency (DC)** | Assesses the model's ability to maintain semantic and referential coherence across multiple turns of clarification. Consistency is measured by tracking dialogue state alignment and the absence of contradictions. |
| **Final Execution Success Rate (FESR)** | Measures whether the final resolved command (after possible clarifications) leads to a safe and correct function execution in the system. This combines successful intent detection, parameter extraction, and ambiguity resolution. |
| **Intent Hit Rate (IHR)** | Evaluates whether the predicted intent label exactly matches the gold-standard intent. This focuses purely on intent-level accuracy independent of parameter filling. |
| **Function Hit Rate (FHR)** | Checks whether the predicted API/function name aligns with the gold-standard function call. This ensures the correct system API is invoked. |
| **Parameter Completeness (F1-Score)** | Measures both the precision and recall of extracted slots/parameters within the predicted function call. F1 balances coverage of required arguments with correctness of extracted values. |
| **Protocol Compliance Rate (PCR)** | Assesses whether generated function calls comply with predefined API schema and safety constraints (e.g., correct slot types, no missing required arguments, no unsafe defaults). High PCR reflects reliability for deployment. |

## C.1 TIER 1: SINGLE-ROUND FUZZY PARSING

This tier evaluates the model's ability to interpret ambiguous single-turn commands with subtle ambiguities (e.g., "Increase the temperature" without a target value). Metrics include:

- **Intent Recognition Accuracy (IRA)**:

$$\text{IRA} = \frac{1}{N} \sum_{i=1}^{N} \mathbb{I}(l_i = \hat{l}_i) \tag{12}$$

- **Parameter Extraction Precision (PEP)**:

$$\text{PEP} = \frac{|\hat{P} \cap P|}{|\hat{P}|} \tag{13}$$

## C.2 TIER 2: EXTREME FUZZY COUNTER-QUESTIONING

This tier tests the model's ability to detect and clarify highly ambiguous commands (e.g., "Turn that switch off"). Metrics include:

- **Fuzzy Detection Rate (FDR)**:

$$\text{FDR} = \frac{\text{TP}}{\text{TP} + \text{FN}} \tag{14}$$

- **Counter-Question Coverage (CQC)**:

$$\text{CQC} = \frac{\sum_{i=1}^{n} \min(m_i, M_i)}{\sum_{i=1}^{n} M_i} \tag{15}$$

## C.3 TIER 3: DYNAMIC MULTI-TURN COMMAND UNDERSTANDING

This tier evaluates context retention and command execution in multi-turn dialogues. Metrics include:

- **Dialogue Consistency (DC)**:

$$\text{DC} = \frac{1}{T} \sum_{t=1}^{T} \cos(s_t, s_{t+1}) \tag{16}$$

- **Final Execution Success Rate (FESR)**:

$$\text{FESR} = \frac{1}{N} \sum_{i=1}^{N} \mathbb{I}(\text{Exec}(\hat{c}_i) = \text{Exec}(c_i)) \tag{17}$$

## C.4 SUPPLEMENTARY INSTRUCTION

**Sensitivity of realism threshold (inverse perplexity)**   We performed a sensitivity analysis on inverse perplexity (IP) realism thresholds across various percentiles in Table 26.

Table 26: Performance metrics under different inverse perplexity thresholds

| Threshold (IP) | Ambiguity ↑ | Protocol ↑ | Realism ↑ | DQS ↑ |
|---|---|---|---|---|
| 5th (strict) | 0.83 | 0.95 | 0.81 | 0.860 |
| 10th | 0.86 | 0.94 | 0.81 | 0.869 |
| 20th (used) | 0.89 | 0.95 | 0.82 | 0.887 |
| 50th (loose) | 0.90 | 0.92 | 0.75 | 0.861 |

- Strict filtering reduces diversity; loose thresholds reduce realism.
- The chosen 20th percentile optimally balances realism, diversity, and adherence.

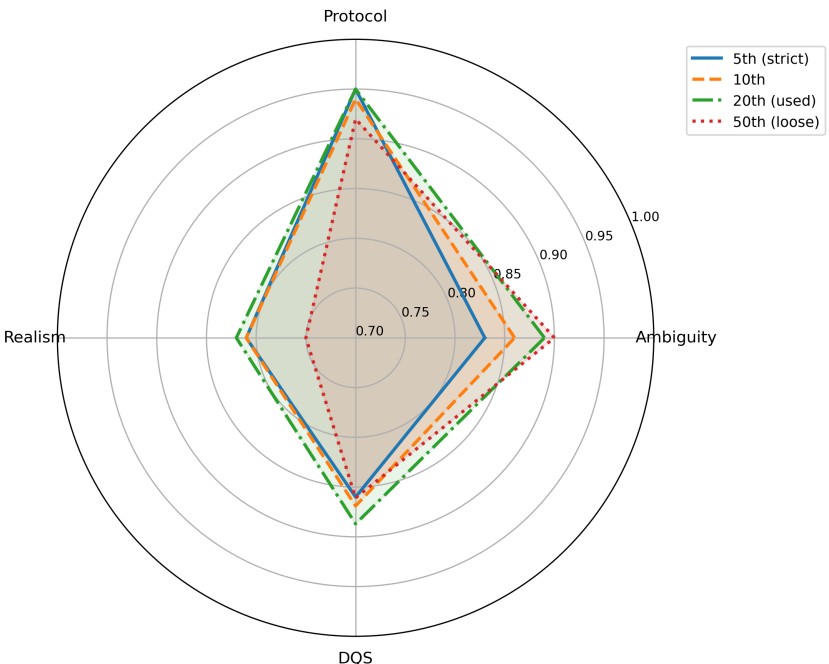

Figure 7: Performance Metrics under Different IP Thresholds

**Rationale for** $\lambda_1 - \lambda_3 = (0.4, 0.3, 0.3)$ **in Eq. (2)** The weight combination $\lambda_1 - \lambda_3 = (0.4, 0.3, 0.3)$ used in Eq. (2) was selected based on careful expert consideration of the domain-specific importance of each evaluation dimension:

- Ambiguity Diversity (AD=0.4): Primary goal to capture diverse ambiguities.

- Protocol Compliance & Realism (PC/R=0.3 each): Essential for validity and authenticity.

Robustness analysis across alternative weights confirms minimal variation in aggregate scores and stable rankings, reinforcing the chosen default (0.4, 0.3, 0.3) configuration in Table 27.

Table 27: Robustness analysis of weight configurations

| Weights (AD, PC, R) | AD ↑ | PC ↑ | R ↑ | DQS(Aggregate Score) ↑ | Rank Change in Table 2 |
|---|---|---|---|---|---|
| (0.40, 0.30, 0.30) Default | 0.89 | 0.95 | 0.82 | 0.887 | - |
| (0.33, 0.33, 0.34) Balanced | 0.89 | 0.95 | 0.82 | 0.886 | No change |
| (0.50, 0.25, 0.25) Emphasis on AD | 0.89 | 0.95 | 0.82 | 0.886 | No change |
| (0.30, 0.50, 0.20) Emphasis on PC | 0.89 | 0.95 | 0.82 | 0.906 | No change |
| (0.20, 0.20, 0.60) Emphasis on R | 0.89 | 0.95 | 0.82 | 0.860 | No change |

**Gold-Standard Function-Call Definition.** For all experiments in Section 4, especially the Function Hit Rate evaluation in Section 4.3, the *gold-standard* structured function call $c_i^*$ for each instruction is determined by a unified validation pipeline tailored to production in-vehicle HMI constraints. A model prediction $\hat{c}_i$ is counted as correct only if it satisfies *all* criteria below.

**Gold standard 1——Semantic Ground-Truth Alignment.** Each instruction is paired with an expert-annotated canonical function call under our unified schema (37 actions, 146 parameters). A prediction must match this call exactly in: (i) action name, (ii) required slot set, (iii) parameter values and types, and (iv) absence of hallucinated arguments.

**Gold standard 2——Schema and Type Validation.** Predicted JSON is checked by a deterministic schema validator that enforces: (i) action–slot legality, (ii) data-type correctness, (iii) value-range constraints (e.g., brightness $\in [0, 100]$), and (iv) mutual-exclusion rules between parameters.

**Gold standard 3——Protocol and Safety Compliance.** Beyond semantic matching, predictions must satisfy production HMI protocol rules, including: (i) state-dependent constraints (e.g., temperature changes allowed only when HVAC is ON), (ii) safety-dependent conditions (e.g., seat heating only if occupancy is detected), (iii) forbidden action–parameter combinations, and (iv) restrictions on overriding locked states (e.g., child-lock windows).

**Gold standard 4——Multi-Step Ordering and Causal Preconditions.** Some instructions trigger multi-step calls (e.g., "navigate home and play music"). Gold-standard sequences additionally satisfy:

- *Functional order compliance:* e.g., start navigation $\rightarrow$ then play music (not reversed).
- *Causal preconditions:* e.g., defogging enabled before adjusting windshield airflow; AC mode compatible with temperature-setting operations.

A prediction violating ordering or causal rules is marked incorrect even if individual calls are syntactically valid.

**Gold standard 5——Real-Log Consistency (for Log-Derived Samples).** For instructions originating from real in-vehicle logs, we additionally verify that predicted calls are consistent with historical execution patterns by retrieving similar utterances from 4M+ anonymized logs via embedding-based search. Similarity thresholds and mappings were reviewed by two OEM experts to ensure real-world alignment.

**Gold standard 6——Human Adjudication for Ambiguous Cases.** For inherently ambiguous or multi-intent instructions ($\sim$12% of the evaluation set), predictions from all models are anonymized, shuffled, and adjudicated by human annotators under a blind protocol. Only predictions consistent with the majority vote are accepted as hits.

**Gold standard 7——Final Decision Rule.** A prediction counts as correct only if all criteria above are satisfied:

$$\text{FHR} = \frac{1}{N} \sum_{i=1}^{N} \mathbb{I}\big[\texttt{AllCriteriaSatisfied}(i)\big].$$

Table 28 summarizes the validation pipeline.

Table 28: Gold-standard validation criteria for function-call evaluation.

| Criterion | Validator | Applies To |
|---|---|---|
| Semantic correctness | Expert annotations | All samples |
| Schema & type validity | JSON schema validator | All samples |
| Protocol & safety compliance | HMI rule engine | All samples |
| Multi-step order & causality | UX/causal rules | Multi-step calls |
| Real-log consistency | 4M-log similarity check | Real-log samples |
| Human majority vote | Blind adjudication | Ambiguous samples |

**Causal relationships analysis in model performance**  Cross-dataset generalization results for both backbones are presented in Table 29, demonstrating the superior performance of ClarifyVC-Data across all evaluation settings.

Across two backbones and three evaluation suites, ClarifyVC-trained models outperform all alternatives by 16–23%. These gains persist across entirely different evaluation distributions and are not tied to log-derived data. They arise from the structure and quality of ClarifyVC-Data rather than model size, training-set scale, or domain overlap. Consequently, the dataset's design is the causal driver of the improvements.

Through controlled ablation, cross-dataset validation, and mechanistic evidence from Table 2 and Figure 3, we show that the observed performance gains are not attributable to distributional overlap, model scale, data quantity, or log-derived exposure bias. Instead, they are causally driven by ClarifyVC's schema-grounded parsing (SPA), adversarial diversity (AGA), ambiguity-focused fuzz injection (FIA), multi-turn recovery supervision (MEA), and the overall difficulty profile of ClarifyVC-Data.

Table 29: External Causal Validation of ClarifyVC-Data: Model Performance Trained on Various Datasets and Evaluated on Unseen, Unbiased Benchmarks

(a) Qwen2.5-7B-Instruct Fine-Tuning Results

| Training Set | Unbiased Test (4k) | Real Logs (2k) | OOD Fuzzy (1k) | Avg |
|---|---|---|---|---|
| **ClarifyVC-Data (20k)** | **92.7%** | **94.1%** | **89.5%** | **92.1%** |
| NL2API-Car | 72.4% | 74.8% | 67.1% | 71.4% |
| Glaive-fc-v2 | 68.3% | 71.0% | 61.5% | 66.9% |
| Talk2Car-FC | 70.2% | 73.6% | 65.7% | 69.8% |
| doScenes-FC | 67.5% | 70.4% | 62.9% | 66.9% |
| Mixed-Open | 75.1% | 76.3% | 69.4% | 73.6% |

(b) LLaMA3-8B-Instruct Fine-Tuning Results

| Training Set | Unbiased Test (4k) | Real Logs (2k) | OOD Fuzzy (1k) | Avg |
|---|---|---|---|---|
| **ClarifyVC-Data (20k)** | **87.9%** | **90.5%** | **85.1%** | **87.8%** |
| NL2API-Car | 63.4% | 66.1% | 58.2% | 62.5% |
| Glaive-fc-v2 | 60.8% | 63.2% | 54.7% | 59.6% |
| Talk2Car-FC | 64.9% | 67.5% | 59.4% | 63.9% |
| doScenes-FC | 61.1% | 63.8% | 56.0% | 60.3% |
| Mixed-Open | 66.7% | 69.0% | 61.5% | 65.7% |

To further validate that these gains are causally tied to the *pipeline-level* design of ClarifyVC-Data—rather than any vehicle-control prior—we additionally apply the full SPA/AGA/FIA/MEA pipeline to three *non-automotive* function-calling datasets: *function-calling-chatml* (general assistant APIs for finance, polling, and utilities), *Glaive-fc-v2* (open-domain task assistant for books, tools, and utilities), and *Arabic_Function_Calling* (multilingual scientific and utility API calls, e.g., boiling/melting points). These corpora differ from in-cabin voice control in schema format, interaction pattern, domain, and linguistic style, and contain no car-related APIs.

In each case, we treat the original corpus as a "base" dataset and use ClarifyVC's SPA/AGA/FIA/MEA pipeline purely as a *data augmentation mechanism*. Importantly, the pipeline only requires: (i) a structured JSON schema, (ii) typed parameters, and (iii) optional safety constraints; it does not encode any automotive-specific bias.

Table 30: DQS improvements on non-automotive function-calling datasets via ClarifyVC pipeline. AD: Ambiguity Diversity; PC: Protocol Compliance; R: Realism.

| Dataset | AD ↑ | PC ↑ | R ↑ |
|---|---|---|---|
| function-calling-chatml (orig) | 0.47 | 0.92 | 0.79 |
| function-calling-chatml (aug) | **0.78** (+0.31) | **0.93** (+0.01) | **0.82** (+0.03) |
| Glaive-fc-v2 (orig) | 0.41 | 0.91 | 0.77 |
| Glaive-fc-v2 (aug) | **0.62** (+0.21) | **0.91** (+0.00) | **0.78** (+0.01) |
| Arabic_Function_Calling (orig) | 0.44 | 0.90 | 0.81 |
| Arabic_Function_Calling (aug) | **0.70** (+0.26) | **0.91** (+0.01) | **0.83** (+0.02) |

As shown in Table 30, ClarifyVC-style augmentation consistently increases Ambiguity Diversity (AD) while keeping Protocol Compliance (PC) effectively unchanged and slightly improving Realism (R). This matches the intended behaviour: the pipeline injects *structured* ambiguity, rather than noise, even when the underlying tools and domains are unrelated to vehicles.

We then fine-tune two backbones (Qwen2.5-7B-Instruct and LLaMA3-8B-Instruct) on either the original or pipeline-augmented versions of these non-automotive corpora, and evaluate all models on a domain-neutral OOD fuzzy benchmark (*OOD Fuzzy 1k*). We report Function Hit Rate (FHR) in Table 31.

Table 31: Function Hit Rate (FHR, %) on a domain-neutral OOD fuzzy benchmark after fine-tuning on non-automotive function-calling corpora, with and without ClarifyVC-style augmentation.

| Model | Training Data | FHR (Orig.) | FHR (Aug.) | Δ |
|---|---|---|---|---|
| Qwen2.5-7B | function-calling-chatml | 72.0 | 88.5 | +16.5 |
| Qwen2.5-7B | Glaive-fc-v2 | 71.4 | 85.2 | +13.8 |
| Qwen2.5-7B | Arabic_Function_Calling | 69.1 | 75.4 | +6.3 |
| LLaMA3-8B | function-calling-chatml | 68.5 | 80.1 | +11.6 |
| LLaMA3-8B | Glaive-fc-v2 | 68.0 | 82.9 | +14.9 |
| LLaMA3-8B | Arabic_Function_Calling | 65.5 | 82.7 | +17.2 |

Across all three non-automotive datasets and both model backbones, ClarifyVC-style augmentation yields consistent FHR gains in the range of +6.3 to +17.2 percentage points, despite the fact that both training and evaluation occur entirely *outside* the vehicle-control domain. Together with the cross-dataset results in Table 29, these experiments provide additional causal evidence that the benefits of ClarifyVC originate from its SPA/AGA/FIA/MEA pipeline and the resulting data difficulty profile.

# D  SUPPLEMENTARY RESULTS

This section provides additional visualizations to complement the results in the main text (Section 4).

## D.1  SUPPLEMENTARY VISUALIZATIONS

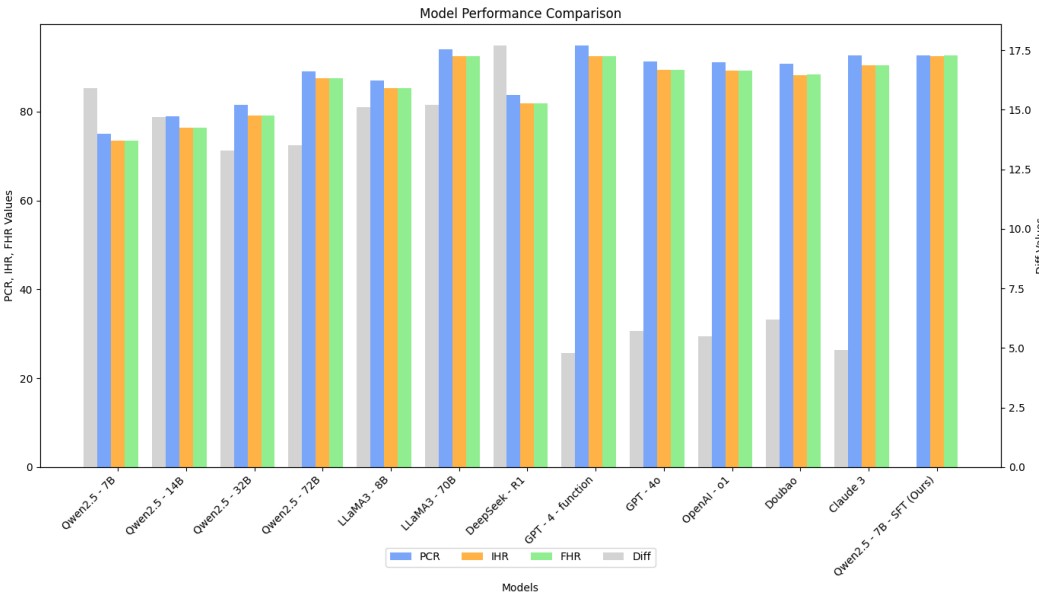

Figure 8: Bar chart comparing model performance across Intent Hit Rate (IHR), Function Hit Rate (FHR), Parameter Completeness (F1-Score), and Protocol Compliance Rate (PCR).

Table 32 reports the zero-shot performance (%) of four large language models—Qwen2.5-72B, LLaMA3-70B, Claude 3, and GPT-4—across five distinct benchmarks. Each benchmark is evaluated on four core metrics: Intent Accuracy, Function Accuracy, Parameter Coverage, and Multi-Turn Success. Overall, GPT-4 consistently achieves the highest scores on all datasets, with the most pronounced advantage observed on the challenging ClarifyVC-Data. The three open-source models show competitive performance on standard tasks such as Talk2Car and CI-AVSR but exhibit substantial drops on APIGen and especially ClarifyVC-Data, highlighting the increased complexity and real-world variability captured by these benchmarks.

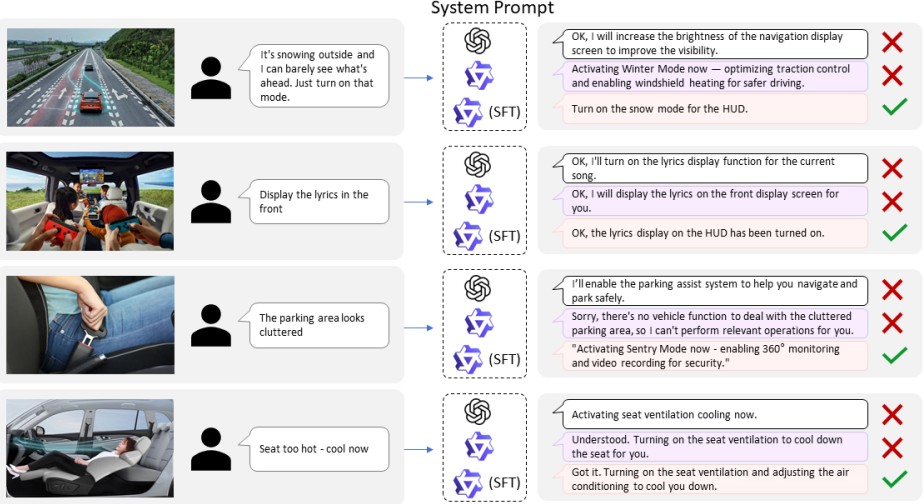

Figure 9: Diagram showing model responses to user queries in various automotive scenarios.

Table 32: Aggregated Zero-Shot Performance (%) of Four LLMs on Five Benchmarks and Four Evaluation Metrics

| Benchmark | Metric | Qwen2.5-72B | LLaMA3-70B | Claude 3 | GPT-4 |
|-----------|--------|-------------|------------|----------|-------|
| Talk2Car | Intent Accuracy | 94.0 | 95.2 | 94.5 | 97.0 |
| | Function Accuracy | 90.1 | 91.3 | 90.7 | 93.9 |
| | Parameter Coverage | 86.6 | 87.9 | 86.8 | 90.5 |
| | Multi-Turn Success | 79.6 | 81.0 | 80.3 | 84.5 |
| CI-AVSR | Intent Accuracy | 90.5 | 92.3 | 91.0 | 94.2 |
| | Function Accuracy | 88.5 | 89.0 | 88.8 | 92.0 |
| | Parameter Coverage | 84.8 | 86.0 | 84.5 | 88.9 |
| | Multi-Turn Success | 77.2 | 78.7 | 77.4 | 82.1 |
| doScenes | Intent Accuracy | 91.5 | 90.8 | 91.7 | 93.5 |
| | Function Accuracy | 89.0 | 87.5 | 89.4 | 91.5 |
| | Parameter Coverage | 85.4 | 84.7 | 85.8 | 88.5 |
| | Multi-Turn Success | 78.4 | 76.8 | 78.8 | 81.5 |
| APIGen | Intent Accuracy | 86.7 | 88.3 | 87.4 | 90.1 |
| | Function Accuracy | 83.2 | 83.8 | 83.9 | 87.2 |
| | Parameter Coverage | 80.2 | 81.0 | 80.5 | 84.8 |
| | Multi-Turn Success | 72.1 | 73.0 | 72.6 | 76.3 |
| ClarifyVC-Data | Intent Accuracy | 72.4 | 74.1 | 73.5 | 77.8 |
| | Function Accuracy | 67.8 | 68.9 | 68.3 | 72.5 |
| | Parameter Coverage | 63.5 | 64.7 | 64.2 | 69.0 |
| | Multi-Turn Success | 62.9 | 63.8 | 63.4 | 67.0 |

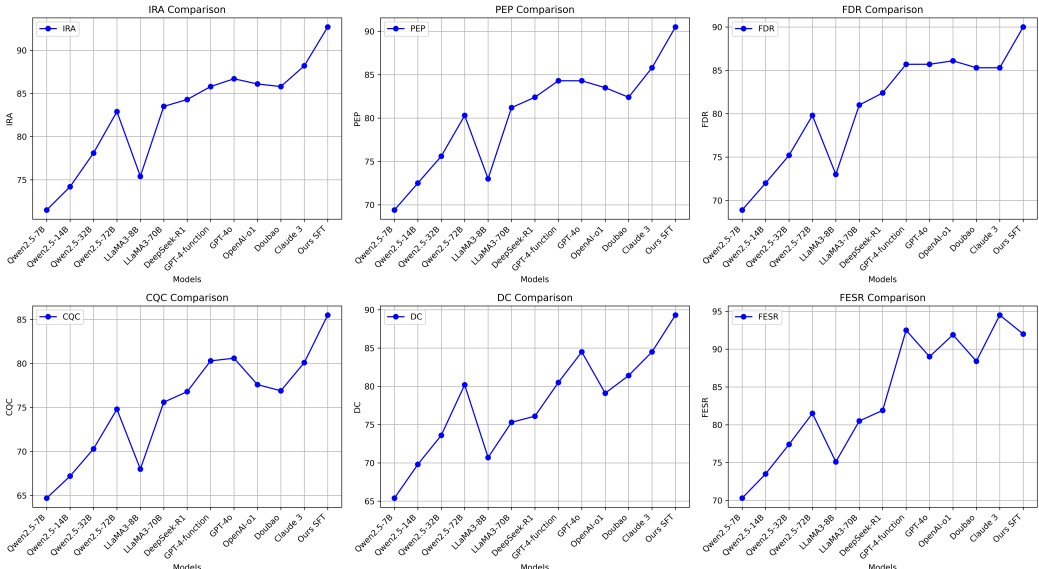

Figure 10: Plot chart illustrating the comprehensive performance profile of different models across key metrics.

The bar chart (Figure 8) highlights the superior performance of the fine-tuned Qwen2.5-7B-SFT model across basic instruction-following metrics. The response diagram (Figure 9) illustrates model behavior in automotive scenarios, while the plot chart (Figure 10) provides a comprehensive performance overview.

Table 33: Advanced scenario evaluation across complex, ambiguous, and multi-turn vehicle-control instructions. Means $\pm$ std over **5** runs; std $< 1\%$ for all SFT rows. Closed-source API models are evaluated once due to usage limits.

| Model | IRA | PEP | FDR | CQC | DC | FESR |
|---|---|---|---|---|---|---|
| Qwen2.5-7B | 71.5 | 69.4 | 68.9 | 64.7 | 65.4 | 70.3 |
| Qwen2.5-14B | 74.2 | 72.5 | 72.0 | 67.2 | 69.8 | 73.5 |
| Qwen2.5-32B | 78.1 | 75.6 | 75.2 | 70.3 | 73.6 | 77.4 |
| Qwen2.5-72B | 82.9 | 80.3 | 79.8 | 74.2 | 78.1 | 81.5 |
| LLaMA3-8B | 75.4 | 73.0 | 72.8 | 68.0 | 70.7 | 75.1 |
| LLaMA3-70B | 83.5 | 81.2 | 81.0 | 75.6 | 80.5 | 83.0 |
| DeepSeek-R1 | 84.3 | 82.4 | 82.0 | 76.8 | 81.4 | 84.1 |
| GPT4-function | 88.5 | 86.2 | 85.7 | 80.3 | 85.0 | 87.8 |
| GPT-4o | 86.7 | 84.3 | 83.9 | 78.4 | 83.0 | 86.0 |
| OpenAI-o1 | 86.1 | 83.5 | 83.0 | 77.6 | 82.2 | 85.2 |
| Doubao | 85.8 | 83.0 | 82.5 | 76.9 | 81.7 | 84.8 |
| Claude 3 | 88.2 | 85.8 | 85.3 | 80.1 | 84.5 | 87.4 |
| **Qwen2.5-7B-SFT (Ours)** | **92.7** $\pm$0.5 | **90.5** $\pm$0.6 | **90.0** $\pm$0.6 | **85.5** $\pm$0.6 | 89.3 $\pm$0.5 | **92.0** $\pm$0.5 |
| Qwen2.5-14B-SFT | 91.8 $\pm$0.4 | 89.7 $\pm$0.5 | 88.9 $\pm$0.6 | 83.8 $\pm$0.6 | 88.1 $\pm$0.5 | 90.1 $\pm$0.5 |
| Qwen2.5-32B-SFT | 92.1 $\pm$0.5 | 90.1 $\pm$0.5 | 89.5 $\pm$0.6 | 84.2 $\pm$0.6 | 89.0 $\pm$0.5 | 90.8 $\pm$0.6 |
| Qwen2.5-72B-SFT | 93.0 $\pm$0.5 | 90.2 $\pm$0.6 | 89.6 $\pm$0.6 | 83.9 $\pm$0.6 | **90.2** $\pm$0.5 | 91.3 $\pm$0.6 |

Table 33 reports advanced scenario evaluation across complex, ambiguous, and multi-turn vehicle control tasks. Baseline open-source and proprietary LLMs show moderate performance: smaller backbones such as Qwen2.5-7B and LLaMA3-8B struggle with fuzzy disambiguation (FDR $< 70$) and long-horizon grounding (FESR $< 75$), while larger backbones (e.g., Qwen2.5-72B, Claude 3) achieve stronger accuracy yet incur high computational overhead. By contrast, our **Qwen2.5-7B-SFT**, fine-tuned on ClarifyVC-Data, consistently outperforms all baselines across six metrics, achieving 92.7 IRA, 90.5 PEP, and 92.0 FESR.

Importantly, although Qwen2.5-72B attains competitive results, the gap between 7B-SFT and 72B is modest ($< 6.6$pp across metrics), while the computational savings are substantial: training costs drop by nearly an order of magnitude and inference latency is reduced $\sim 10\times$, making 7B-SFT far more practical for deployment in resource-constrained, safety-critical environments. These results demonstrate that targeted exposure to ambiguity-rich yet schema-compliant supervision substantially improves semantic parsing, safe clarification, and multi-turn grounding, yielding models that are both accurate and deployment-efficient compared to significantly larger backbones.

## LIMITATIONS

While **ClarifyVC-Data** advances the evaluation of function call understanding in vehicle command scenarios, several limitations remain:

- **Modality Scope.** Our benchmark primarily focuses on text-based instruction understanding. Although future vehicle systems often involve multimodal contexts (e.g., vision, LiDAR, spatial audio), these are not yet fully integrated into the current benchmark version. Extending ClarifyVC to multimodal grounding is an important future direction, and we are actively developing a multimodal variant to enable such research.

- **Domain Generalizability.** Although the function-call schema is designed to be extensible, current task templates are oriented toward the in-cabin control setting. Extending the dataset to cover broader domains such as driving policy, diagnostics, or V2X communication would improve general applicability.

- **Evaluation Reliance on Static Metrics.** Our proposed metrics (e.g., IRA, FDR, CQC) evaluate alignment and robustness in a static fashion. However, real-time interaction and downstream driving consequences (e.g., safety violations) are not yet modeled in the evaluation pipeline.

- **Language Biases.** As the current benchmark is constructed in English, it may not generalize across linguistic or cultural variations in vehicle command phrasing. Future work can consider multilingual and dialectical command variants.

Despite these limitations, we believe ClarifyVC-Data lays a critical foundation for robust benchmarking in vehicle-focused LLM deployments and opens pathways for future expansion in modality, task complexity, and real-world grounding.

