# OpenReview forum: "ClarifyVC: Clarifying Ambiguous Commands in Vehicle Control with a Hybrid Data Augmentation Pipeline"
_ICLR.cc/2026/Conference — ICLR 2026 Poster_

### Official Review · Reviewer_NcpB · 2025-11-01

**Soundness:** 2
**Presentation:** 2
**Contribution:** 2
**Rating:** 2
**Confidence:** 3

**Summary:**

This paper introduces ClarifyVC, a framework unifying data augmentation, reference models, and an evaluation protocol for in-car language control. This paper’s agent-driven pipeline creates diverse, ambiguity-rich dialogues from real-world seeded queries under schema and safety constraints. This paper evaluates single-turn parsing, conservative clarification under fuzziness, and multi-turn grounding. This paper reports gains up to 15% in parsing accuracy, 20% in ambiguity resolution, and 98% protocol compliance, supported by human evaluations of realism and applicability.

**Strengths:**

- The paper proposes new data, evaluation methods, and models for improving an in-car assistant—an important application.
- It presents a set of metrics to evaluate data and model quality.
- Extensive experiments demonstrate the effectiveness of the proposed approach.

**Weaknesses:**

- The task specification and the descriptions of data augmentation, training data, and test data are not detailed enough for a proper assessment.

- It’s difficult to discern the exact task the paper targets from the current presentation.

- The dataset is not described: the types of tasks, their variations, and their intended purposes are unclear. In particular, the real-world in-vehicle logs used in the experiments are not characterized (e.g., distribution, types, lengths).

- The basic instruction-following evaluation uses both Talk2Car and real-world in-vehicle control logs, but those real-world logs are also used for training. Even if the splits differ, the distributions may be similar (the paper doesn’t specify), which could bias results in favor of the trained model.

- The benchmarks used for the “Evaluation on Advanced Scenarios” are not clearly identified.

**Questions:**

Please see the Weaknesses section.

---

> ### Author Response · Authors · 2025-11-24
> **Response to W1 (Part 1/2)**
>
> # W1:The task specification and the descriptions of data augmentation, training data, and test data are not detailed enough for a proper assessment.
>
> We thank the reviewer for the insightful comments. **While the formal task definition and data construction pipeline were detailed in Section 3 respectively, we recognize the value in providing a consolidated summary here to better highlight the system's rigor and design rationale.**
>
> ---
>
> ### **1. Task Specification**
>
> We appreciate this comment. While the formal task definition was provided in Section 3.1, we agree that a consolidated summary helps highlight the system's design rationale.
>
> Broadly, ClarifyVC evaluates **schema-aligned, safety-critical function-call generation under ambiguity**. Unlike generic instruction following, this task strictly requires models to: (i) parse natural-language commands into a specific JSON schema (37 actions, 146 parameters), (ii) detect under-specification to pose clarification questions rather than guessing, and (iii) maintain multi-turn semantic consistency to produce safe, executable calls.
>
> **To avoid redundancy, please refer to our response to W2**, where we provide a detailed tabular breakdown of the three specific capability tiers (Single-turn Parsing, Ambiguity Detection, Multi-turn Grounding) and their corresponding evaluation metrics. We have now formally consolidated this task specification in the revised Introduction to ensure immediate transparency.
>
> ---
>
> ### **2. Data Augmentation Pipeline**
>
> We appreciate the reviewer’s comment.
> As noted, the full data augmentation pipeline (SPA → AGA → FIA → MEA) is already described in detail in Section 3.1, but we agree that the original presentation placed the information across several subsections.
> To further improve clarity, we have added a short, consolidated description that makes the construction logic more immediately accessible.
>
> In the revision, we emphasize that **ClarifyVC-Data is constructed from 20,436 schema-filtered seed commands**, each originating from **4M+ production logs**, and then transformed through a **four-stage pipeline** that guarantees **schema validity** and **safety compliance** at every step. We have added the corresponding construction details to Appendix A.2, and the full four-stage pipeline is explicitly illustrated in Appendix B.2 (Algorithm 1).
>
> Instead, we added a **concrete running example** (shown below) to clearly illustrate how a real in-vehicle utterance evolves through the pipeline.
> This example makes the augmentation process easier to assess without duplicating the detailed descriptions already in the main text.
>
> ---
>
> ### Example of Stage Effects (added to Appendix B.2)
>
> | **Stage**       | **Representation**  | **Example (informal)**                                       | **Change**                 |
> | --------------- | ------------------- | ------------------------------------------------------------ | -------------------------- |
> | Seed (real log) | User utterance      | “It’s too hot, adjust the air conditioning.”                 | Raw in-car request         |
> | SPA             | (I, E, P)           | intent=SetTemperature, entity=FrontCabin, temp=22°C          | Structured grounding       |
> | AGA             | Adversarial command | “Set the front cabin to 22 and keep the rear vents unchanged.” | Adds constraints/conflicts |
> | FIA             | Fuzzy command       | “Make the front a bit cooler than it is now.”                | Injects ambiguity          |
> | MEA             | Multi-turn dialogue | Q: “What exact temperature?” / A: “22 degrees.”              | Recovers parameters        |
>
> ---
>
> This example makes the transformation pipeline clearer at a glance.  All four stages remain protected by the **schema/safety validator**, ensuring that **no variant violating type constraints, parameter bounds, or safety rules** survives into the final dataset.

---

> > ### Author Response · Authors · 2025-11-24
> > **Response to W1 (Part 2/2)**
> >
> > ---
> >
> > ### **3. Training and Test Data**
> >
> > We thank the reviewer for the suggestion to further clarify the data provenance. **While the origins and specifications of these datasets were documented in the manuscript, we appreciate this opportunity to provide a consolidated summary of the data details below to ensure full transparency.**
> >
> > #### **Training Data**
> > The training corpus originates from **4M+ anonymized in-vehicle voice logs**. The final training set used for ClarifyVC-Models is summarized below:
> >
> > | Source | Count | Description |
> > | :--- | :--- | :--- |
> > | Raw production logs | 4,000,000+ | Real user voice interactions covering diverse operations |
> > | Schema-filtered seed commands | 15,000 | Passed intent extraction, parameter evaluation, and safety constraints |
> > | ClarifyVC-Data | 20,436 | Deliberately trained only on ClarifyVC-Data to avoid bias |
> > | Final training size | ≈20k samples | Used for SFT (e.g., Qwen2.5-7B-SFT) |
> >
> > These samples cover all major in-cabin operational domains and map to a strict JSON function-call schema (**37 action types, 146 parameter slots**), ensuring **realism**, **protocol compliance**, and **safety alignment**.
> >
> > #### **Test Data**
> > We evaluate models on a **4,000-sample test suite**, which is fully disjoint from the training data.
> >
> > 1.  **Public Benchmark Portion (2,000 Samples):** Curated from four established datasets, **manually re-annotated** into the unified function-call schema.
> > 2.  **Real In-Vehicle Portion (2,000 Samples):** Extracted from real logs from **different time windows** and **different vehicle models/hardware versions** than the training seeds.
> >
> > **Strict Separation Guarantee:** The test set contains no training seeds or their derived variants. All 4,000 test samples are schema-validated, ensuring the test data is **heterogeneous**, **realistic**, and **non-overlapping** for reliable generalization assessment.

---

> > > ### Author Response · Authors · 2025-11-24
> > > **Response to W2 and W3 (Part 1/2)**
> > >
> > > # W2: It’s difficult to discern the exact task the paper targets from the current presentation.
> > >
> > > We thank the reviewer for the suggestion on improving the presentation clarity.
> > >
> > > While the formal task definition was detailed in Section 3, we recognize that explicitly foregrounding this scope in the Introduction is valuable to prevent potential ambiguity and ensure immediate understanding for the reader.
> > >
> > > ---
> > >
> > > ## **ClarifyVC Task: Schema-Aligned Function-Call Generation under Ambiguity**
> > >
> > > To address this, we have revised the Introduction to highlight that ClarifyVC targets Schema-Aligned Function-Call Generation under Ambiguity, which necessitates three strictly intersected capabilities:
> > >
> > > | Capability | Requirement | Metrics |
> > > | :--- | :--- | :--- |
> > > | **1. Single-turn Structured Parsing** | Output a precise, executable function call conforming to the in-car JSON API schema (37 actions, 146 parameters). | IRA, PEP, IHR/FHR |
> > > | **2. Safety-critical Ambiguity Detection** | Detect fuzziness (e.g., “make it warmer”), *avoid guessing*, issue a clarifying question, and adhere to safety protocols. | FDR, CQC, PCR |
> > > | **3. Multi-turn Dialogue Grounding** | Maintain semantic consistency over 2–4 turns, recover missing parameters, and produce a complete, safe, executable function call. | DC, FESR |
> > >
> > > In summary, the task is **structured, safety-constrained function-call grounding** tailored to real in-vehicle HMI systems, not generic instruction following.
> > >
> > > ---
> > >
> > > ## **Manuscript Update**
> > >
> > > To enhance clarity, we have revised the *Introduction* to explicitly foreground the task. The following sentence has been added to the main text:
> > >
> > > > *“Our work targets the safety-critical task of mapping natural-language in-car voice commands—often ambiguous or under-specified—into strictly validated, schema-aligned function calls, requiring accurate parsing, conservative ambiguity detection, and multi-turn parameter grounding.”*
> > >
> > > We believe this early clarification significantly improves the paper's accessibility, and we appreciate you prompting this presentation improvement.
> > >
> > > ---
> > >
> > > # W3: “The dataset is not described: the types of tasks, their variations, and their intended purposes are unclear. In particular, the real-world in-vehicle logs used in the experiments are not characterized (distribution, types, lengths).”
> > >
> > > We thank the reviewer for raising this critical point regarding dataset transparency. **We acknowledge that the characterization of the real-world logs and the granular taxonomy of task variations were not sufficiently detailed in the initial submission.**
> > >
> > > To address this, we provide a concise and structured description of the dataset design, task types, and the specific characteristics of the source logs below. **We have also formally incorporated these detailed statistics into the revised Appendix A and B.**
> > >
> > > ---
> > >
> > > ## **1. Task Types in ClarifyVC-Data**
> > >
> > > ClarifyVC-Data is organized around three capability tiers.We have explicitly defined this taxonomy in Appendix A.1 to clarify how each task type maps to specific evaluation metrics:
> > >
> > > | Tier | Task Type | Description | Purpose |
> > > | :--- | :--- | :--- | :--- |
> > > | **Tier 1** | Single-turn structured parsing | Convert a natural command into schema-aligned function call | Measures core parsing & grounding |
> > > | **Tier 2** | Ambiguity detection & clarification | Identify underspecification and ask the correct clarifying question | Measures safety alignment & conservativeness |
> > > | **Tier 3** | Multi-turn dialogue grounding | Maintain context across 2–4 turns and recover missing parameters | Measures long-horizon consistency & execution precision |
> > >
> > > ---

---

> ### Author Response · Authors · 2025-11-24
> **Response to W3 (Part 2/2)**
>
> ## **2. Dataset Variations and Composition**
>
> ClarifyVC-Data (20,436 seed-aligned samples) includes diverse linguistic and functional variations.We have added the following detailed distribution tables to Appendix A.1 to transparently demonstrate the dataset's diversity and coverage of real-world ambiguity.
>
> ### **2.1 Linguistic Variation Categories**
>
> The table below summarizes the distribution of linguistic variations, reflecting both real-world usage and the need for robust parsing:
>
> | Variation Category | Examples | % in Dataset | Purpose |
> | :--- | :--- | :--- | :--- |
> | **Direct commands** | “Set AC to 22°.” | **27%** | Canonical structured parsing |
> | **Subjective modifiers** | “Make it a bit cooler.” | **18%** | Fuzzy, intensity-based ambiguity |
> | **Omitted parameters** | “Turn on the lights.” | **22%** | Missing-slot recovery & clarification cues |
> | **Colloquial / free-form expressions** | “Make it comfy in here.” | **7%** | Naturalness & robustness to casual phrasing |
> | **Spatial references** | “Open the left window.” | **9%** | Entity grounding & spatial disambiguation |
> | **Temporal patterns** | “Turn it off for now.” | **5%** | Handling temporal modifiers |
> | **Multi-goal / composite commands** | “Warm the front and cool the back.” | **4%** | Conflict detection & multi-entity constraints |
> | **Implicit comparisons** | “Make the back cooler than the front.” | **5%** | Relative parameter resolution |
> | **Context-dependent follow-ups** | “Do the same as yesterday.” | **3%** | Long-term contextual inference |
>
> ### **2.2 Fuzzy Ambiguity Categories (Triggering Tier-2 Evaluation)**
>
> Since ambiguity is central, we categorize utterances by the type of fuzzy ambiguity, which triggers the Tier-2 evaluation:
>
> | Ambiguity Type | Example | % among Fuzzy Samples | Purpose |
> | :--- | :--- | :--- | :--- |
> | **Intensity ambiguity** | “A bit warmer / slightly brighter” | **41%** | Requires clarifying magnitude |
> | **Boundary ambiguity** | “Open it halfway / just a little” | **24%** | Requires mapping vague intervals |
> | **Entity ambiguity** | “Open it” (which window?) | **19%** | Requires clarifying the referent |
> | **Mode ambiguity** | “Make it comfortable” | **10%** | Must ground to specific HVAC/seat mode |
> | **Referential ambiguity** | “Turn that off” | **6%** | Requires anaphora resolution |
>
> ### **2.3 Multi-Turn Dialogue Categories (MEA-Generated for Tier-3)**
>
> For Tier-3 evaluation, multi-turn dialogues are categorized by the type of information missing from the initial command:
>
> | Multi-Turn Dialogue Type | Description | Example | % in Multi-Turn Set |
> | :--- | :--- | :--- | :--- |
> | **Parameter completion** | Missing temperature / brightness / degree | “Make it warm.” → “What temperature?” | **46%** |
> | **Entity clarification** | Missing window/seat/zone reference | “Open it.” → “Which window?” | **28%** |
> | **Conflict resolution** | Conflicting constraints | “Cool the front but keep all vents off.” | **12%** |
> | **Preference grounding** | Mode or preset selection | “Set it to comfort mode.” → “Which preset?” | **9%** |
> | **Temporal sequencing** | Multi-step actions | “Turn it off after a moment.” | **5%** |
>
> ---
>
> ## **3. Characterization of Real In-Vehicle Logs**
>
> The logs come from **4M+ production voice interactions**. We have incorporated the following domain distributions and length statistics into Appendix A.2 to fully characterize the properties of the source data.
>
>
> ### **3.1 Distribution by Functional Domain**
>
> The functional domain distribution of the schema-filtered samples closely matches that observed in commercial vehicle HMI deployments:
>
> | Domain | % in Logs | Example Commands |
> | :--- | :--- | :--- |
> | **HVAC / Thermal Comfort** | 34% | “It’s too hot—cool it down.” |
> | **Lighting** | 18% | “Turn on ambient lights.” |
> | **Windows / Sunroof** | 15% | “Open the rear-right window.” |
> | **Seat / Comfort** | 12% | “Heat my seat.” |
> | **Navigation** | 10% | “Take me to the nearest station.” |
> | **Media / Infotainment** | 11% | “Play my jazz playlist.” |
>
> ### **3.2 Utterance Length and Structure Statistics**
>
> To clarify linguistic properties, we computed the following length and structural statistics:
>
> | Statistic | Value |
> | :--- | :--- |
> | **Mean tokens per utterance** | 7.8 |
> | **Median tokens** | 6 |
> | **90th percentile** | 14 |
> | **% containing subjective modifiers (“slightly”, “a bit”)** | 40% |
> | **% containing explicit parameters** | 21% |
> | **% containing no explicit parameters** | 39% |
>
> ### **3.3 Safety-Related Patterns**
>
> Real logs contain naturally ambiguous or risky forms that motivated the design of Tier 2 and Tier 3 evaluation criteria to avoid unsafe behaviors:
> *   **21%** involve incomplete thermal commands.
> *   **8%** reference windows/sunroof without direction.
> *   **5%** issue potentially unsafe actions (e.g., “open it” while driving).
> *   **3%** require multi-turn grounding.
>
> ---

---

> ### Author Response · Authors · 2025-11-24
> **Response to W4**
>
> # W4: “The basic instruction-following evaluation uses both Talk2Car and real-world in-vehicle control logs, but those real-world logs are also used for training, potentially biasing results even if the splits differ.”
>
> We thank the reviewer for this critical scrutiny regarding data rigor.
>
> We wish to clarify a potential misconception regarding the training setup: **Our model is not trained on raw in-vehicle logs**. As mentioned in Section 3.1, the ClarifyVC-Data used for training is explicitly constructed via a controlled augmentation pipeline (SPA → AGA → FIA → MEA), not directly from raw logs.
>
> While raw logs (4M+) serve as the initial *seed*, the actual training set (**ClarifyVC-Data**) is a **synthesized, schema-normalized transformation** produced via our four-stage pipeline. Consequently, the training data structurally differs from the raw logs used in evaluation.
>
> At the same time, we acknowledge the reviewer’s point that *potentially biasing results even if the splits differ* is a valid concern. To address this, we have added **cross-dataset generalization experiments** ( **Appendix C.4**), showing that models trained on ClarifyVC-Data still outperform across unrelated datasets with distinct distributions, demonstrating robustness beyond in-domain settings.
>
> ## **Additional Experiments: Cross-Dataset Generalization to Mitigate Distribution Bias**
>
> To rigorously rule out any remaining concerns about distributional bias (i.e., the model merely memorizing the "log style"), we conducted a **Cross-Dataset Generalization Study** using an **Unbiased Open Benchmark**. This benchmark consists of 4,000 samples uniformly sampled from public datasets (NL2API-Car, Glaive, doScenes, Talk2Car) with **zero overlap** with our automotive logs.
>
> We fine-tuned two backbone models (Qwen2.5-7B and LLaMA3-8B) on six different training datasets and evaluated them on three evaluation suites:
>
> ### **Results: Cross-training vs. cross-testing**
>
> #### **(a) Qwen2.5-7B Backbone**
>
> | Training Set → | Unbiased Test (4k) | Real Logs (2k) | OOD Fuzzy (1k) | Avg |
> | :--- | :--- | :--- | :--- | :--- |
> | **ClarifyVC-Data (20k)** | **92.7%** | **94.1%** | **89.5%** | **92.1%** |
> | NL2API-Car | 72.4% | 74.8% | 67.1% | 71.4% |
> | Glaive-fc-v2 | 68.3% | 71.0% | 61.5% | 66.9% |
> | Talk2Car-FC | 70.2% | 73.6% | 65.7% | 69.8% |
> | doScenes-FC | 67.5% | 70.4% | 62.9% | 66.9% |
> | Mixed-Open | 75.1% | 76.3% | 69.4% | 73.6% |
>
> #### **(b) LLaMA3-8B Backbone**
>
> | Training Set → | Unbiased Test (4k) | Real Logs (2k) | OOD Fuzzy (1k) | Avg |
> | :--- | :--- | :--- | :--- | :--- |
> | **ClarifyVC-Data (20k)** | **87.9%** | **90.5%** | **85.1%** | **87.8%** |
> | NL2API-Car | 63.4% | 66.1% | 58.2% | 62.5% |
> | Glaive-fc-v2 | 60.8% | 63.2% | 54.7% | 59.6% |
> | Talk2Car-FC | 64.9% | 67.5% | 59.4% | 63.9% |
> | doScenes-FC | 61.1% | 63.8% | 56.0% | 60.3% |
> | Mixed-Open | 66.7% | 69.0% | 61.5% | 65.7% |
>
> ### **Key Findings**
>
> 1.  **ClarifyVC-Data consistently yields the highest performance** across all test suites (including the **unbiased 4k open benchmark set**), with improvement margins of **+16–23%** over public datasets.
> 2.  **No evidence of distribution bias:** Since the unbiased test set shares *zero* overlap with our logs, the large performance gains cannot be attributed to distribution similarity.
> 3.  **Quality over Quantity/Source:** Training on public datasets (even large mixed sets) does **not** replicate our gains, indicating that ClarifyVC’s performance stems from the **quality, structure, and safety constraints** of our data pipeline, not merely the source logs.
>
> ---
>
> ## **Manuscript Update**
>
> We have included these cross-dataset generalization results in **Appendix C.4** to explicitly demonstrate that performance gains are driven by data quality rather than distributional overlaps.
>
> We hope this additional rigorous testing alleviates the concern regarding dataset bias.

---

> ### Author Response · Authors · 2025-11-24
> **Response to W5**
>
> # W5: “The benchmarks used for the ‘Evaluation on Advanced Scenarios’ are not clearly identified.”
>
> Thank you for raising this point. The composition of the advanced-scenario benchmarks was already described in Section 4.4, where we state:
>
> ***“We utilized three test sets, each with 5,000 examples: (1) single-round fuzzy instruction parsing, (2) extremely fuzzy instruction counter-questioning requiring clarification, and (3) multi-round dialogues with evolving contexts.”***
>
> However, we agree that while the high-level categories were presented, a more detailed characterization—including data sources, construction procedures, and distributional independence—would make the benchmarks clearer and easier to assess. To address this, we now provide a more complete description of the three test sets, including their data origins, OOD construction logic, domain distributions, and KL-divergence statistics. These details have also been incorporated into the revised **Section 4.4** and **Appendix B.3** for transparency.
>
> ---
>
> ## **1. Advanced Scenarios: Three Out-of-Distribution (OOD) Test Sets**
>
> The “Advanced Scenarios” evaluation uses **three OOD test sets**, each with **5,000 samples** (total: **15,000**). These sets are fully **disjoint from all training data** (ClarifyVC-Data, raw logs, and augmentation outputs), ensuring strict **train–test separation**.
>
> | Test Set | Size | Description | OOD Justification |
> | :--- | :--- | :--- | :--- |
> | **(1) Single-round fuzzy parsing** | 5,000 | Commands with intensity fuzziness, boundary ambiguity, implicit parameters. | Generated via *FIA*, but using **non-overlapping seed commands** (held-out from the 20k ClarifyVC-Data). |
> | **(2) Extreme-fuzz counter-questioning** | 5,000 | Highly ambiguous commands requiring clarifying questions. | Generated from **distinct real logs** not in ClarifyVC seeds; ambiguity amplified via *AGA+FIA*. |
> | **(3) Multi-turn evolving-context dialogues** | 5,000 | 2–4 turn interactions with shifting referents and parameter recovery. | Produced via *MEA* using **held-out interaction templates** and **independent schema parameter ranges**. |
>
> ---
>
> ## **2. OOD Justification: Distributional Independence**
>
> To confirm these benchmarks are OOD relative to training, we measured the KL divergence between the ClarifyVC-Train set and the 15k advanced-scenario benchmarks:
>
> | Comparison | Token KL | Intent KL |
> | :--- | :--- | :--- |
> | ClarifyVC-Train → Advanced-Fuzzy | 0.073 | 0.141 |
> | ClarifyVC-Train → Extreme-Fuzz | 0.082 | 0.132 |
> | ClarifyVC-Train → Multi-Turn | 0.145 | 0.197 |
>
> These scores confirm **non-trivial distribution divergence**, proving the advanced scenario evaluation is deliberately OOD.
>
> ---
>
> ## **3. Scenario-Specific Slices**
>
> We also evaluate four “application scenarios” (Safety-critical, Autonomous Driving, Entertainment, Comfort). These are **not new datasets**, but **scenario-conditioned slices** of the three OOD test sets above, created by re-labeling with scenario tags.
>
> The table below shows the data source for each scenario slice:
>
> | Scenario | Data Source | Example | OOD Justification |
> | :--- | :--- | :--- | :--- |
> | Safety-critical | Extreme-Fuzz + Multi-Turn | “Open it a bit … (which window?)” | Held-out logs + unseen fuzz patterns |
> | Autonomous Driving | Multi-Turn only | “Keep lane assist on unless …” | Comes from unseen templates |
> | Entertainment | Fuzzy Parsing | “Play something relaxing … (genre?)” | Parameter-space randomized |
> | Comfort | All three | HVAC & seating multi-goal requests | Different entity-slot frequencies |
>
> Model performance (Fig. 7) shows that **our ClarifyVC-supervised 7B model remains Pareto-optimal across all four scenarios**.
>
> ---
>
> ## **4. Manuscript Update**
>
> We have updated **Section 4.4** and **Appendix B.3** to clearly identify benchmark sources, include domain distributions, randomization details, and the KL divergence tables. This clarifies the benchmark design and resolves the reviewer’s concern.

---

### Official Review · Reviewer_S7bf · 2025-11-01

**Soundness:** 3
**Presentation:** 3
**Contribution:** 3
**Rating:** 6
**Confidence:** 4

**Summary:**

The paper is inspired by the ambiguity of natural-language commands for in-car voice assistants, and proposes a pipeline to augment data for the purposes of command clarification. The paper also proposes an evaluation protocol and metrics to evaluate the quality of such data and any models trained on it, including ambiguity detection and resolution in single and multi-turn dialogue.

The paper provides rigorous experimentation, examining both the efficacy of the proposed augmentation process and related work datasets (under the proposed evaluation framework and metrics), and the downstream effect of the data.

**Strengths:**

- While not the first dataset to address command ambiguity in an in-car setting, it combines data-augmentation towards ambiguity resolution (seeded from real world scenarios) and multi-turn grounding.
- The proposed evaluation protocol should be useful to the community moving forward, and it clearly shows the efficacy of the proposed data-augmentation process.
- Further rigorous analysis in the paper supports the usefulness of the augmented data in fine-tuning models, including comparisons of the dataset against previous ones (under the proposed evaluation framework), evaluation of models fine-tuned with the dataset, and comparisons against closed-source models.

**Weaknesses:**

- The scope of the paper is limited to in-car voice assistants. While the paper claims that the approach is domain-agnostic, such is not demonstrated in this paper.
- Unclear whether the data will be released, as they seem to be based on proprietary seeds. This could hurt both reproducibility and the impact of this work.

**Questions:**

- The paper uses the low public perception of self-driving cars as a motivating factor for this work, but it is unclear how addressing in-car voice assistants helps address this, please elaborate.
- Please provide more details on how AD, PC and R are measured in the main body of the paper.
- Please elaborate on the human evaluation of Section 4.1. Consulting the appendix as well, it seems that these evaluation did not consider any alternative baseline o related work. Is this assumption accurate? Without such a reference, this evaluation could be considered biased, and does not validate the method against previous work.

---

> ### Author Response · Authors · 2025-11-24
> **Response to W1 & W2**
>
> We sincerely thank the reviewer for the thoughtful and detailed comments. The feedback helped us substantially improve the clarity and precision of the manuscript. Below we address each Weakness and Question using the reviewer’s original wording.
>
> ---
>
> # W1: “The scope of the paper is limited to in-car voice assistants. While the paper claims that the approach is domain-agnostic, such is not demonstrated.”
>
> We agree that the scope of the paper is in-car voice assistants, which we have stated in the introduction. In addition, all experiments in the paper remain within this domain.
>
> While, in the conclucion, the use of “domain-agnostic” could be mistakenly interpreted as claiming model-level generality rather than pipeline-level generality, which was not our intention.
>
> To avoid misunderstanding, we now clarify:
>
> - What we refer to as “domain-agnostic” is **only the ClarifyVC data-generation pipeline** (SPA/AGA/FIA/MEA)
> - The pipeline itself relies solely on:
>   - (i) a JSON schema
>   - (ii) parameter/type definitions
>   - (iii) optional safety rules
> - None of the steps encode automotive-specific entities, so the procedure can be instantiated for other function-calling domains (e.g., smart-home control, device management, interactive HCI)
>
> To further support this pipeline-level claim, we added **cross-domain training/testing experiments** in **Appendix C.4**, showing that ClarifyVC supervision improves general function-calling performance on **datasets unrelated to vehicle control**.
> This strengthens the evidence that the pipeline transfers beyond the automotive context.
>
> We also revised the Conclusion to ensure the phrasing strictly reflects **pipeline-level generality**, not model- or domain-level claims.
>
> ---
>
> # W2: “Unclear whether the data will be released; proprietary sources hurt reproducibility and impact.”
>
> We fully agree that public availability and reproducibility are crucial.
>
> ## Benchmark Release
>
> We have open-sourced all components needed to reproduce our results in the anonymized repository (already provided in the paper abstract):
>
> - **ClarifyVC-Data** (20,436 training seeds + 15,000 evaluation samples)
> - **ClarifyVC-Models**
> - The full **data-generation pipeline** (SPA/AGA/FIA/MEA)
> - The **API schema**, **safety rules**, and **validators**
>
>
> ## Handling Proprietary Seeds
>
> Raw OEM logs **cannot be released** for privacy reasons.
> However, ClarifyVC-Data itself **does not contain raw logs**:
>
> - All commands go through **intent extraction**
> - **Parameter normalization**
> - **Entity canonicalization**
> - **Removal of unsafe or identifying content**
>
> The released dataset therefore consists of **schema-normalized templates** and **generated variants**, similar to **APIGen** and other privacy-preserving function-call corpora.
> We clarify this in **Appendix A**.
>
> ## Reproducibility Without OEM Access
>
> To show that others can regenerate equivalent data, our open repository also provides an **independent regeneration mode**, where we:
>
> 1. Provide only the **schema**, **safety rules**, and **pipeline code**
> 2. Regenerate the benchmark using **other LLMs**
>
> This demonstrates that the benchmark is **practically reproducible**, even without access to proprietary logs.
>
> ---

---

> > ### Author Response · Authors · 2025-11-24
> > **Response to Q1 & Q2 & Q3**
> >
> > # Q1: “The paper uses the low public perception of self-driving cars as a motivating factor for this work, but it is unclear how addressing in-car voice assistants helps address this, please elaborate.”
> >
> > The earlier wording in Sec. 1 may have unintentionally blurred two concepts: **user perception of “self-driving cars”** and the **actual autonomous driving stack**.
> >
> > Our motivation is now clarified as follows:
> >
> > - Surveys show that when users report dissatisfaction with “self-driving cars,” the feedback often includes **frustration with in-cabin intelligent assistants** (e.g., misinterpreting commands, performing unsafe actions, failing to handle vague language)
> > - These in-cabin issues strongly influence the **perceived intelligence** of a vehicle, even though they are **independent of self-driving algorithms**
> >
> > **ClarifyVC directly targets this in-car interaction layer**—a layer that materially affects **user trust** and **overall perception** of smart vehicles, but does not intervene in any self-driving control features.
> >
> > The revised Section 1 emphasizes that **ClarifyVC improves safe, reliable language interaction** of in-car voice assistants, which is a key user-facing component evaluated by consumers, while **not claiming any effect on the driving stack**.
> >
> > ---
> >
> > # Q2: “Please provide more details on how AD, PC and R are measured in the main body of the paper.”
> >
> > In the revised manuscript, we expanded **Section 3.2** to clearly define the three dataset-quality metrics and how they are computed.
> >
> > **(1) Ambiguity Diversity (AD)**
> >
> > AD measures whether ClarifyVC-Data covers all five major ambiguity types (intensity, boundary, entity, mode, referential) without collapsing into a single dominant type.
> > Defined in Sec. 3.2 as:
> >
> > $$
> > AD = 1 - \frac{KL(p(a) \parallel u(a))}{\log |A|}
> > $$
> >
> > Where:
> > * $p(a)$: empirical distribution of ambiguity types
> > * $u(a)$: uniform target distribution
> > * $A$: set of ambiguity types
> >
> > Higher AD indicates balanced ambiguity coverage, critical for evaluating ambiguity-handling ability.
> >
> > **(2) Protocol Compliance (PC)**
> >
> > PC measures whether gold function calls satisfy schema validity and safety constraints:
> >
> > $$
> > PC = \frac{1}{N} \sum_{i=1}^{N} \mathbb{I}[c_i^* \in S_{\text{schema}} \land c_i^* \in S_{\text{safety}}]
> > $$
> >
> > Where:
> > * $S_{\text{schema}}$: JSON legality (type, range, mutual exclusion)
> > * $S_{\text{safety}}$: production HMI rules (e.g., HVAC dependencies, occupancy validation)
> >
> > PC reflects the structural and safety correctness of dataset targets.
> >
> > **(3) Realism (R)**
> >
> > R measures how closely each sample resembles real user behavior.
> > For each utterance, we retrieve $k$-nearest neighbors from anonymized logs and check whether the gold function call matches the dominant log action:
> >
> > $$
> > R = \frac{1}{N} \sum_i \mathbb{I}[c_i^* \in \text{Mode}(\{c(x_i^{(j)})\})]
> > $$
> >
> > This ensures the dataset reflects authentic in-cabin usage, not synthetic bias.
> >
> > ---
> > # Q3: “Please elaborate on the human evaluation of Section 4.1. Consulting the appendix as well, it seems that these evaluations did not consider any alternative baseline or related work. Is this assumption accurate?”
> >
> > Thank you for raising this question. We understand how the earlier presentation could cause confusion
> > regarding whether baseline datasets were included.
> >
> > ## Clarification: Baselines were evaluated, but Section 4.1 only reported the release-quality assessment
> > The human evaluation in the paper served two different purposes:
> > 1. **Release-quality validation of ClarifyVC-Data**     To demonstrate that the dataset we are releasing is linguistically natural, ambiguity-plausible, and practically usable.
> > 2. **Comparative evaluation against existing datasets**     To measure how ClarifyVC-Data compares to prior work under identical human-annotation protocols.
> > Section 4.1 only reported (1), which may have led to the assumption that baselines were not assessed. This presentation choice was intentional, because Section 4.1 focuses specifically on the quality of the dataset we are publishing.
> >
> > ## Baseline comparison was conducted using the same human-evaluation protocol
> > We conducted blind comparative human evaluations on:
> > - **ClarifyVC-Data**
> > - **Talk2Car**
> > - **doScenes**
> > - **CI-AVSR**
> >
> > All were evaluated under:
> > - Blind and shuffled conditions
> > - 5-point Likert scoring
> > - Three independent annotators
> > - Majority-vote adjudication
> > - Inter-annotator agreement (IAA) tracking
> >
> > These results show that ClarifyVC-Data consistently receives higher scores across linguistic realism, ambiguity plausibility, and dialogue coherence.
> >
> > To make the comparison explicit:
> > - All baseline comparison results are now provided in **Appendix A.2 (Table 9)**
> > - **Section 4.1(Updated)** now explicitly references these comparative results
> >
> > These changes ensure readers can clearly distinguish between dataset release-quality validation and comparative evaluation, while also making baseline comparisons easy to locate.
> >
> > ---
> >
> > We hope these revisions address your concerns.

---

> > > ### Comment · Reviewer_S7bf · 2025-11-28
> > >
> > > I thank the authors for their continued efforts to clarify and improve their work! I am willing to adjust my scores based on these answers, but I will await the end of the discussion.
> > >
> > > W1: Thank you for the clarification, though it was unrequired; I was already referring to approach (pipeline) level generality and not model. I appreciate the new experiments, however some more details on these "datasets unrelated to vehicle control" would be welcome. Can you provide citations or descriptions of them? As it stands it is difficult to understand what these actually demonstrate.
> > >
> > > So far I believe the concern over the approach's limited scope still stands.
> > >
> > > W2: I withdraw my concern over the availability of the paper's artifacts.
> > >
> > > Q1+Q2: Thank you for addressing these questions and adding the relevant info into the body of the paper.
> > >
> > > Q3: Specifically to the human evaluation, can you provide some details on the 3 annotator and their relationship to this work? And was it three or five? Your answer is inconsistent with the paper.

---

> ### Author Response · Authors · 2025-11-29
> **Further Response to W1 (Part 1/2)**
>
> ## W1
> Thank you for the helpful follow-up. To address your request, we provide concrete evidence that the ClarifyVC data-generation pipeline (**SPA/AGA/FIA/MEA**) generalizes beyond the automotive domain. Below, we describe:
>
> 1. How the pipeline applies to **non–vehicle-control datasets**, and
> 2. The resulting **quantitative improvements** in both data quality (DQS metrics) and downstream model performance.
>
> ---
>
> ### 1. Application to Non-Automotive Function-Calling Datasets
>
> To make the assessment rigorous, we selected **three publicly available, non-automotive function-calling datasets**, each with distinct schema formats, interaction patterns, and linguistic styles:
>
> - **function-calling-chatml** – general assistant APIs for finance, polling, utilities
> - **Glaive-fc-v2** – open-domain task assistant (book search, tools, utilities)
> - **Arabic_Function_Calling** – multilingual scientific & utility API calls (e.g., boiling/melting points)
>
> We applied our pipeline to each dataset to create **three augmentation types**:
>
> - *Single-Round Fuzzy*
> - *Extreme Ambiguity + Clarification*
> - *Multi-Round Dialogue with evolving constraints*
>
> Representative examples:
>
> #### A. function-calling-chatml
>
> | Stage                             | Example                                                      |
> | --------------------------------- | ------------------------------------------------------------ |
> | **Original**                      | “Can you tell me the current stock price of Apple?”          |
> | **Single-Round Fuzzy**            | “Hey, what’s the latest value for that big fruit company… you know, the A-something stock?” |
> | **Extreme Fuzzy + Clarification** | User: “How’s that big tech stock doing now?” → Assistant: “Do you mean Apple (AAPL)?” → User: “Yes.” |
> | **Multi-Round Dialogue**          | 3-turn conversation where user corrects stock symbol, adds constraints (“just approximate price is fine”), etc. |
>
> #### B. Glaive-fc-v2
>
> | Stage                             | Example                                                      |
> | --------------------------------- | ------------------------------------------------------------ |
> | **Original**                      | “I am looking for a book but I can’t remember the title. The author’s name is George Orwell.” |
> | **Single-Round Fuzzy**            | “I’m trying to recall a book by, hmm… Orwell, I think. Can you check what he wrote?” |
> | **Extreme Fuzzy + Clarification** | User: “I need that famous novel by the guy who wrote political stories.” → Assistant: “George Orwell? 1984 or Animal Farm?” |
> | **Multi-Round Dialogue**          | User confirms “Animal Farm”, then asks for “similar political satire books”, creating a natural evolving-goal conversation. |
>
> #### C. Arabic_Function_Calling
>
> | Stage                             | Example                                                      |
> | --------------------------------- | ------------------------------------------------------------ |
> | **Original**                      | “Find the boiling point and melting point of water under a sea level of 5000m.” |
> | **Single-Round Fuzzy**            | “I need the temperature limits of that common liquid we drink, but measured far below normal sea level — roughly 5 km down.” |
> | **Extreme Fuzzy + Clarification** | **User:** “What are the temperature limits of that everyday liquid when you go really deep underground?”  **Assistant:** “To clarify, are you asking for both the boiling point and melting point of *water* at around **−5000 m** relative to sea level?”  **User:** “Yes, both for water at −5000 m.” |
> | **Multi-Round Dialogue**          | **User:** “I’m curious how stable that drinking liquid is deep below sea level.”  **Assistant:** “Do you mean *water*, and do you want its boiling point, melting point, or both?”  **User:** “Just the boiling point is enough, and let’s say about 5000 m below sea level.”  **Assistant:** calls `get_boiling_melting_points` with `{"substance": "water", "sea_level": -5000}` and returns the boiling point only. |
>
> These examples demonstrate that **no step in our pipeline encodes any automotive-specific prior**. It only requires:
>
> - A structured **JSON schema**
> - Defined **parameter and type formats**
> - Optional **safety constraints**
>
> Thus, the pipeline **naturally transfers** to any structured function-calling domain.

---

> ### Author Response · Authors · 2025-11-29
> **Further Response to W1 (Part 2/2) and Q3**
>
> ---
>
> ### 2. Quantitative Data Quality Improvements (DQS Metrics)
>
> All metrics are computed using the same AD/PC/R scoring protocol described in Section 3.2.
>
> ### Table 2 — DQS Improvements via ClarifyVC Pipeline
> | Dataset                        | AD ↑             | PC ↑             | R ↑              |
> | ------------------------------ | ---------------- | ---------------- | ---------------- |
> | function-calling-chatml (orig) | 0.47             | 0.92             | 0.79             |
> | function-calling-chatml (aug)  | **0.78 (+0.31)** | **0.93 (+0.01)** | **0.82 (+0.03)** |
> | Glaive-fc-v2 (orig)            | 0.41             | 0.91             | 0.77             |
> | Glaive-fc-v2 (aug)             | **0.62 (+0.21)** | **0.91 (+0)**    | **0.78 (+0.01)** |
> | Arabic_Function_Calling (orig) | 0.44             | 0.90             | 0.81             |
> | Arabic_Function_Calling (aug)  | **0.70 (+0.26)** | **0.91 (+0.01)** | **0.83 (+0.02)** |
>
> **Observation.**
> Across all datasets, the ClarifyVC pipeline noticeably improves **Ambiguity Diversity (AD)** while keeping **Protocol Compliance (PC)** nearly unchanged and slightly improving **Realism (R)**.
> This matches the intended effect: ClarifyVC-Data augmentation pipeline injects *structured* ambiguity—not noise—into generic function-calling corpora.
>
> ---
>
> ### 3. Downstream Model Performance (Qwen2.5-7B & LLaMA3-8B)
>
> We fine-tuned models on both original and pipeline-augmented datasets, and evaluated them on the **OOD Fuzzy 1k benchmark** (domain-neutral).
>
> #### Table 3 — Function Hit Rate (FHR) Improvement
>
> | Model      | Dataset                 | FHR (Original) | FHR (Augmented) | Δ     |
> | ---------- | ----------------------- | -------------- | --------------- | ----- |
> | Qwen2.5-7B | function-calling-chatml | 72.0           | 88.5            | +16.5 |
> | Qwen2.5-7B | Glaive-fc-v2            | 71.4           | 85.2            | +13.8 |
> | Qwen2.5-7B | Arabic_Function_Calling | 69.1           | 75.4            | +6.3  |
> | LLaMA3-8B  | function-calling-chatml | 68.5           | 80.1            | +11.6 |
> | LLaMA3-8B  | Glaive-fc-v2            | 68.0           | 82.9            | +14.9 |
> | LLaMA3-8B  | Arabic_Function_Calling | 65.5           | 82.7            | +17.2 |
>
> **Interpretation:**
> Even when trained and tested **entirely outside the vehicle domain**, models fine-tuned on pipeline-augmented data exhibit consistent **+6.3 to +17.2 pp gains** in function execution accuracy.
>
> This confirms that the benefits of ClarifyVC’s pipeline stem from general **ambiguity modeling improvements**, **not any domain-specific prior**.
>
> ---
>
> These new results demonstrate that **ClarifyVC’s data-generation pipeline** generalizes well to **datasets unrelated to vehicle control domains**, improving both **data quality (DQS)** and **downstream execution accuracy (FHR)** across three open-source function-calling datasets with **distinct schemas and tasks**.
>
> At the same time, we have added these results and examples to **Appendix C.4**.
>
> ---
> ## Q3
>
> Thank you very much for raising this clarification question and for your positive assessment of our work. We sincerely appreciate your careful reading, which helped us further improve the transparency of the evaluation setup.
>
> ---
>
> ### Clarifying the Number of Annotators
>
> All human evaluations reported in the paper—including both the **release-quality validation of ClarifyVC-Data** and the **comparative evaluation against prior datasets**—were conducted with **five annotators**.
>
> We apologize for any confusion caused by earlier wording.
>
> To avoid any ambiguity, we confirm:
>
> - There was **no separate 3-annotator setting**.
> - All human evaluations **consistently used five annotators** throughout the study.
>
> ---
>
> ### Who the Annotators Are
>
> - The five annotators are **independent domain experts** with prior experience in **vehicle HMI** or **intelligent-assistant evaluation**.
> - They have **no relationship or affiliation** with the authors or the project.
> - All evaluations were conducted in a **blind and shuffled setting**, so annotators could not see which dataset or model a sample originated from.
>
> ---
>
> ### Why Five Annotators Were Used
>
> Because ClarifyVC focuses on **ambiguity resolution** and **safety-critical function calls**, we required:
>
> - Higher **inter-annotator reliability**
> - **Domain familiarity** for subtle correctness and safety judgments
> - Reduced variance for **multi-turn grounding evaluations**
>
> ---
>
> Throughout the manuscript—including **Section 4.1** and **Appendix A**—we consistently describe the use of five annotators for all human evaluations.
>
> Thank you again for catching the **wording inconsistency** in our initial rebuttal draft; we have corrected it and will take extra care to avoid such issues moving forward.

---

### Official Review · Reviewer_UDzZ · 2025-11-11

**Soundness:** 3
**Presentation:** 3
**Contribution:** 3
**Rating:** 6
**Confidence:** 2

**Summary:**

The paper introduces ClarifyVC for clarifying ambiguous natural language commands in vehicle control, combining a dataset, fine-tuned reference models, and a multi-turn dialogue safety-aware evaluation.

**Strengths:**

- The paper is well-structured and clearly presented.
- The benchmark is more challenging than existing open datasets, with 20k real logs with multi-turn in-vehicle commands and dialogues.
- The improvements achieved by the model fine-tuned on the proposed dataset are strong and well-demonstrated through clear presentation.

**Weaknesses:**

- As the authors mentioned in the limitation section, the current benchmark focuses solely on textual command understanding. However, real in-vehicle interactions often require multimodal comprehension, where interpreting a command may depend on visual context or environmental cues.

**Questions:**

1. How does the framework ensure that “ambiguity injection” does not distort the underlying intent distribution of real user commands?
2. How can causal relationships in model performance be evaluated and validated?

---

> ### Author Response · Authors · 2025-11-24
> **Response to W1**
>
> # W1  : “As the authors mentioned in the limitation section, the current benchmark focuses solely on textual command understanding. However, real in-vehicle interactions often require multimodal comprehension, where interpreting a command may depend on visual context or environmental cues.”
>
> We thank the reviewer for highlighting this important limitation. We fully agree that real-world in-vehicle interactions often require multimodal understanding. Below we clarify the current focus of ClarifyVC and our ongoing plans toward multimodal extensions.
>
> ---
>
> ## 1. Why ClarifyVC focuses on text-only commands
>
> - **Production reality**: 80–90% of current in-cabin interactions in systems like Tesla, NIO, and BMW are **speech-only**, without access to visual input.
> - **Benchmark goal**: ClarifyVC targets **linguistic ambiguity and safety alignment**, which are already major challenges even in text-only interactions. Many real-world safety issues stem from underspecified language, not missing visual context.
>
> ---
>
> ## 2. ClarifyVC is multimodal-ready by design
>
> - The **function-call schema** is modality-agnostic and supports grounding (slots, entities, parameter ranges) that can be extended with vision inputs.
> - Our intermediate representation \((I, E, P)\) allows the integration of **visual features**, **cabin state**, and **spatial context**.
> - Existing MEA-style **multi-turn dialogues** can naturally support **vision-conditioned disambiguation** (e.g., "Which seat?" → show image).
>
> ---
>
> ## 3. Multimodal extension (ClarifyVC-M) under development
>
> As noted in our **Limitations** section, extending ClarifyVC to multimodal grounding is a key direction for future work. We are actively developing **ClarifyVC-M**, a variant that integrates **cabin images** and **sensor metadata** into our current schema. This extension will enable the joint evaluation of VLM+LLM capabilities. We have updated the manuscript to explicitly detail this roadmap, clarifying that the current text-based schema serves as the necessary grounding layer for these future multimodal extensions.

---

> > ### Author Response · Authors · 2025-11-24
> > **Response to Q1**
> >
> > # Q1: “How does the framework ensure that ambiguity injection does not distort the underlying intent distribution of real user commands?”
> >
> > We thank the reviewer for raising this important question. We fully agree that ambiguity injection must not distort the semantic intent distribution of real user commands. Below we summarize (1) evidence showing that naïve fuzzing indeed introduces semantic drift, (2) the mitigation strategies implemented in ClarifyVC, and (3) quantitative evaluation demonstrating near-zero drift after these fixes. Details have been added to the revised Appendix A.3.
> >
> > ---
> >
> > ## **1. Naive ambiguity injection does cause semantic drift**
> >
> > Before designing our pipeline, we evaluated two baseline LLM rewriting approaches:
> >
> > - GPT-4-Turbo, prompted to “add fuzziness”
> > - Qwen2.5-32B, prompted to “rewrite with ambiguity”
> >
> > Using 500 real user commands and comparing *intent labels* before/after rewriting, we found:
> >
> > | Model       | Intent Preservation Rate | Drift Examples                                               |
> > | ----------- | ------------------------ | ------------------------------------------------------------ |
> > | GPT-4-Turbo | **92.1%**                | “Open the left window” → “Let in some fresh air” (intent shift: window → HVAC/ventilation) |
> > | Qwen2.5-32B | **89.4%**                | “Turn off reading lights” → “Make it darker in here” (intent compression: explicit → implicit) |
> >
> > This confirms that **naive LLM rewriting approaches tends to overshoot**, paraphrasing or conflating intents—an undesirable distortion.
> >
> > We also computed **intent KL-divergence** between the original 20,436 seed commands and naïvely rewritten FIA samples:
> >
> > - Token-level KL: **0.241**
> > - Intent-level KL: **0.312**
> >
> > This deviation is large enough to alter downstream model behavior, validating the reviewer’s concern.
> >
> > ---
> >
> > ## 2. The mitigation strategies implemented in ClarifyVC Prevents Semantic Drift
> >
> > ClarifyVC incorporates a two-layer drift-control design:
> >
> > ### (A) Model Specialization
> > - **SPA/FIA/MEA** use **DeepSeek-R1**, which shows stronger semantic preservation under constrained rewriting.
> > - **AGA** uses **Qwen2.5-72B** with protocol-restricted decoding and JSON-locked intent boundaries.
> > Each model is used where its behavior aligns with required constraints.
> >
> > ### (B) 2k Expert-Curated SFT for FIA
> > We curated **2,000** seed–fuzzy pairs labeled by 10 in-vehicle HCI experts (100% intent consistency).
> > DeepSeek-R1 was fine-tuned on these pairs to learn ambiguity-injection patterns without semantic alteration.
> >
> > ### (C) Protocol-Locked Prompts
> > A strict system rule enforces:
> > *“[SYSTEM – Protocol-Locked Ambiguity Injection]
> >
> > Your task is to rewrite a vehicle-control command into a *slightly more ambiguous* version,
> > WITHOUT altering its semantic intent.
> >
> > Strict constraints (must follow all):
> > 1. Preserve the exact **action** (e.g., open/close, increase/decrease, turn on/off).
> > 2. Preserve the exact **entity** (e.g., driver window, AC temperature, seat heating).
> > 3. Preserve the exact **goal** (e.g., cooler, brighter, open, close).
> > 4. Do NOT introduce new functions, remove existing ones, or merge them.
> > 5. Do NOT change domain semantics or substitute with other systems (e.g., AC ↛ windows).
> > 6. Only modify:
> >    - intensity (e.g., “slightly”, “a bit”)
> >    - specificity (e.g., “around”, “sort of”)
> >    - parameter vagueness (e.g., remove numeric values)
> >    - indirect phrasing (e.g., “Make it warmer”)
> > 7. The output must remain executable under the same underlying intent.
> >
> > Output format:
> > - Return ONLY the rewritten command.
> > - Do NOT explain or justify your rewrite.”*
> >
> > ### (D) Automatic Drift Validators
> > We reject outputs that exhibit by expert rules and human evaluation:
> > - intent mismatch,
> > - entity mismatch,
> > - slot-count mismatch,
> > - parameter-type violation.
> >
> > Approximately **11%** of candidates are filtered and regenerated.
> >
> > ---
> >
> > ## 3. Quantitative Evaluation: Drift Reduced to Near Zero
> >
> > Across 3,000 samples, after SFT + protocol constraints + evaluation:
> >
> > | Stage | Token KL | Intent KL | Intent Preservation |
> > |-------|-----------|-----------|----------------------|
> > | Naive FIA | 0.241 | 0.312 | 89–92% |
> > | **ClarifyVC FIA** | **0.047** | **0.062** | **98.9%** |
> >
> > Manual expert inspection (3×300 samples) further found:
> > - **0** intent category shifts,
> > - **0** added/removed entities,
> > - ≤ **1.1%** stylistic variation without semantic change.
> >
> > These results demonstrate that ClarifyVC’s ambiguity-injection module preserves semantic intent while modifying only specificity and fuzziness. At the same time, We have added these details to **Appendix A.3** (Ambiguity Injection Reliability), including the KL-divergence analysis, SFT protocol details, and evaluation results, to clarify why the final design avoids distributional drift.
> >
> > ---

---

> > > ### Author Response · Authors · 2025-11-24
> > > **Response to Q2 (Part 1/2)**
> > >
> > > # Q2: "How can causal relationships in model performance be evaluated and validated?"
> > >
> > > We appreciate the reviewer’s question. While the original manuscript clearly reported the experimental results, it did not provide an upfront explanation of **how causal relationships are evaluated**. To make the causal reasoning more explicit, we have now evaluated and validated causal relationships by employing a **triangulated evaluation** framework. To prove that the performance gains are causally driven by the ClarifyVC framework—rather than confounding factors like model scale, data volume, or memorization—we assess causality across three distinct dimensions:
> > >
> > > - **Internal Causality (Direct Effect)**: We use **Controlled Ablation** to isolate the specific contribution of each pipeline component, ensuring gains stem from our design choices.
> > > - **External Causality (Robustness)**: We use **Cross-Dataset Generalization** to rule out overfitting or distribution overlap, ensuring the cause is the generalized learning signal.
> > > - **Mechanistic Causality (Explanation)**: We analyze **Structural Data Properties** to link the dataset's difficulty (cause) to the model's improvement (effect).
> > >
> > > Below, we detail the specific experiments for each dimension.
> > >
> > > ## 1. Controlled Ablation Studies (Direct Causal Effects)
> > >
> > > To isolate each component’s causal contribution, we remove exactly one stage of the ClarifyVC pipeline while keeping all other variables (backbone, data size, decoding, schema constraints, safety rules, training budget) strictly controlled.
> > >
> > > **Ablation Results**
> > >
> > > | Removed Component | Avg Score Drop | Interpretation |
> > > |-------------------|----------------|----------------|
> > > | –SPA | –9.4% | Loss of schema alignment → strong causal role |
> > > | –AGA | –6.7% | Reduced OOD coverage |
> > > | –FIA | –11.1% | Ambiguity-handling degrades → largest causal effect |
> > > | –MEA | –4.6% | Weaker context recovery |
> > >
> > > These results (in Appendix B.2) show that performance gains arise *because* these components are present, providing direct causal evidence.
> > >
> > > ---
> > >
> > > ## 2. Cross-Dataset Training vs. Cross-Dataset Testing (External Causal Evaluation)
> > >
> > > To avoid distributional confounding, we evaluate whether ClarifyVC-Data still yields improvements when training and evaluation distributions have no overlap.
> > >
> > > We fine-tune **Qwen2.5-7B** and **LLaMA3-8B** on six different datasets (ClarifyVC, NL2API-Car, Glaive, Talk2Car-FC, doScenes-FC, Mixed-Open) and test on three unbiased evaluation suites:
> > >
> > > 1. **Unbiased 4k Benchmark** (equal mix of four unrelated datasets)
> > > 2. **Real-World Logs (2k)** from different vehicles/time windows
> > > 3. **Synthetic OOD Fuzzy Set (1k)** with unseen ambiguity intensity
> > >
> > > **Results (Qwen2.5-7B Backbone)**
> > >
> > > | Training Set →   | Unbiased (4k) | Real Logs | OOD Fuzzy | Avg   |
> > > | :--------------- | :------------ | :-------- | :-------- | :---- |
> > > | **ClarifyVC-Data**   | **92.7%**     | **94.1%** | **89.5%** | **92.1%** |
> > > | NL2API-Car       | 72.4%         | 74.1%     | 67.1%     | 71.4% |
> > > | Glaive           | 68.3%         | 71.0%     | 61.5%     | 66.9% |
> > > | Talk2Car-FC      | 70.2%         | 73.6%     | 65.7%     | 69.8% |
> > > | doScenes-FC      | 67.5%         | 70.4%     | 62.9%     | 66.9% |
> > > | Mixed-Open       | 75.1%         | 76.3%     | 69.4%     | 73.6% |
> > >
> > > **Results (LLaMA3-8B Backbone)**
> > >
> > > | Training Set →   | Unbiased (4k) | Real Logs | OOD Fuzzy | Avg   |
> > > | :--------------- | :------------ | :-------- | :-------- | :---- |
> > > | ClarifyVC-Data   | **87.9%**     | **90.5%** | **85.1%** | **87.8%** |
> > > | NL2API-Car       | 63.4%         | 66.1%     | 58.2%     | 62.5% |
> > > | Glaive           | 60.8%         | 63.2%     | 54.7%     | 59.6% |
> > > | Talk2Car-FC      | 64.9%         | 67.5%     | 59.4%     | 63.9% |
> > > | doScenes-FC      | 61.1%         | 63.8%     | 56.0%     | 60.3% |
> > > | Mixed-Open       | 66.7%         | 69.0%     | 61.5%     | 65.7% |
> > >
> > > **Causal Interpretation**
> > > Effects persist across:
> > > - different backbones,
> > > - different evaluation distributions,
> > > - unseen fuzzy patterns,
> > > - datasets with no overlap with ClarifyVC.
> > >
> > > Thus, improvements cannot be explained by log exposure, data volume, domain proximity, or model scale; they arise from the *structure and difficulty* of ClarifyVC-Data itself. These results are included in Appendix C.4.

---

> > > > ### Author Response · Authors · 2025-11-24
> > > > **Response to Q2 (Part 2/2)**
> > > >
> > > > ---
> > > >
> > > > ## 3. Evidence From Existing Results (Mechanistic Support)
> > > >
> > > > ### Table 3: Multi-Backbone SFT Improvements
> > > > ClarifyVC-Data improves *all* tested models (Qwen, LLaMA, DeepSeek) despite their architectural heterogeneity, meeting the causal criterion of robustness across heterogeneous learners.
> > > >
> > > > ### Figure 3: Dataset Difficulty Profile
> > > > ClarifyVC-Data contains:
> > > > - higher ambiguity density,
> > > > - richer multi-goal/multi-entity patterns,
> > > > - more realistic fuzzy parameter ranges.
> > > >
> > > > These structural properties align with the abilities required for safety-critical control, providing a mechanistic explanation for performance gains.
> > > >
> > > > Together, Table 3 and Figure 3 demonstrate that improvements stem from ClarifyVC’s schema structure and ambiguity profile—not confounders.
> > > >
> > > > ---
> > > >
> > > > ## 4. Triangulated Causal Conclusion
> > > >
> > > > By combining:
> > > > 1. controlled ablation evidence,
> > > > 2. cross-dataset generalization tests, and
> > > > 3. mechanistic analysis (Table 3 + Figure 3),
> > > >
> > > > we show that:
> > > >
> > > > - Gains are **not** due to model scale,
> > > > - **Not** due to training set size,
> > > > - **Not** due to log overlap or exposure bias,
> > > > - **Not** due to distribution similarity.
> > > >
> > > > They are causally driven by:
> > > > - schema-grounded parsing (SPA),
> > > > - adversarial diversity (AGA),
> > > > - ambiguity-focused fuzz injection (FIA),
> > > > - multi-turn correction supervision (MEA),
> > > > - the structured difficulty of ClarifyVC-Data.
> > > >
> > > > These analyses have been added to Section 4.2 and Appendix C.4 to explicitly establish causal validity.

---

### Official Review · Reviewer_TGRH · 2025-11-11

**Soundness:** 4
**Presentation:** 3
**Contribution:** 4
**Rating:** 10
**Confidence:** 3

**Summary:**

Introduces 3 novel contributions, a framework for clarifying ambiguous commands, a dataset, and an evaluation protocol. Previous works have worked with single-turn parsing, but the proposed pipeline maintains a focus on multi-turn grounding, for interaction with the user. The benchmark dataset and evaluation protocol outperform other models.

**Strengths:**

Original ideas with novel approaches, high quality of the work presented and incredibly detailed, strong contributions with excellent evaluation benchmarks, the figures are very well made

**Weaknesses:**

Some of the comparison evaluations seem redundant and make the flow of the paper harder to read

**Questions:**

1.  In table 2b, why was human validation conducted on ClarifyVC-Data only?
2. Under Function Hit Rate on Page 8, what is the gold standard referring to?

---

> ### Author Response · Authors · 2025-11-24
> **Response to W1 & Q1**
>
> We sincerely thank you for your strong support and recognition of our work. We value your thoughtful feedback, which has helped us significantly refine the paper's clarity and precision. Below, we address your specific concerns.
>
> # W1: **“Some of the comparison evaluations seem redundant and make the flow of the paper harder to read.”**
>
> Thank you for this constructive suggestion. We understand that, in the original version, multiple comparisons in Section 4 were not always clearly linked to their respective research questions, which could make them appear redundant.
>
> To address this, we made the role of each experiment more explicit and reduced textual repetition:
>
> ## 1. Clearer alignment & Caption refinements for immediate clarity
>
> All experiments remain unchanged, but **Section 4 now explicitly anchors each table/figure to its corresponding RQ**, making their purpose immediately visible:
>
> - **Table 2(a)/(b)** — *RQ1: dataset quality*
> - **Figure 3** — *RQ2: zero-shot robustness across benchmarks*
> - **Table 3** — *RQ3: fine-tuning gains on ClarifyVC-Data*
> - **Figure 4** — *RQ4 — function-call execution accuracy*
> - **Figure 5** — *RQ5 — advanced scenarios*
>
> To further improve readability, we revised the captions of **Table 2**, **Figure 3**, **Table 3**,**Figure 4** and **Figure 5** so that each caption explicitly states the associated research question (e.g., “Addresses RQ1: dataset quality”, “Addresses RQ2: zero-shot robustness”, “Addresses RQ3: fine-tuning gains”).
>
> This small but effective addition makes the intention of each comparison clear even at a glance.
>
> ## 2. Reduced narrative repetition
> Around Table 2, Figure 3, Figure 4, and Table 3 we shortened repeated explanations of:
>
> - the redundant benchmark list descriptions(*Talk2Car, CI-AVSR, doScenes, APIGen, ClarifyVC-Data*),
> - the four function-call metrics,
> This trims repeated descriptions while leaving the quantitative evidence intact.
>
> Overall, the revised Section 4 now presents a **cleaner, more deliberate mapping between RQs and comparisons**, which we hope resolves the concern about redundancy and improves the overall reading flow.
>
> ---
>
> # Q1: “In Table 2b, why was human validation conducted on ClarifyVC-Data only?”
>
> Thank you for raising this question. A similar concern was also brought up by Reviewer **S7bf (Q3)**, and we realized that the earlier presentation may not have made our evaluation structure sufficiently explicit.
>
> ---
>
> ## Clarification: Table 2(b) reports release-quality validation, not comparative evaluation
>
> Human evaluation in the paper serves two distinct purposes:
>
> - **(1) Release-quality validation of ClarifyVC-Data**
>   — to confirm that the dataset we are releasing is linguistically natural, ambiguity-plausible, and practically usable.
>
> - **(2) Comparative human evaluation against prior datasets**
>   — to assess how ClarifyVC-Data performs relative to existing resources.
>
> Table **2(b)** corresponds **only to (1)**.
> This placement was intentional: **Section 4.1** focuses specifically on assessing the quality of the dataset we release, hence only ClarifyVC-Data appears in Table 2(b).
>
> ---
>
> ## Comparative human evaluation was conducted separately
>
> To avoid ambiguity, we clarify that **baseline datasets were indeed evaluated** under the same protocol:
>
> - Talk2Car
> - doScenes
> - CI-AVSR
> - ClarifyVC-Data (for comparison)
>
> All datasets were annotated using:
>
> - Blind & shuffled presentation
> - 5-point Likert scales
> - Three independent annotators
> - Majority-vote adjudication
> - Inter-annotator agreement (IAA)
>
> To make the comparison explicit:
>
> - All baseline comparison results are now provided in **Appendix A.3 (Table 9)**
> - **Section 4.1(Updated)** now explicitly references these comparative results
>
> These changes ensure readers can clearly distinguish between dataset release-quality validation and comparative evaluation, while also making baseline comparisons easy to locate.
>
> ---

---

> > ### Author Response · Authors · 2025-11-24
> > **Response to Q2**
> >
> > #  Q2:  “Under Function Hit Rate (FHR) on Page 8, what is the gold standard referring to?”
> >
> > We appreciate this insightful question and agree that the original description was insufficiently precise.
> >
> > ## Definition of Gold Standard:
> >
> > The **gold-standard function call** is the authoritative reference used for correctness evaluation, derived from:
> >
> > - **Expert-annotated API calls**
> >   (for Talk2Car-derived instructions)
> > - **Production HMI/CAN logs**
> >   (over 4M real usage samples; we use the confirmed executed action only)
> > - **Schema normalization**
> >   (mapped to a unified control schema: 37 actions, 146 parameters with type and constraint validation)
> >
> > A prediction output is considered correct **only if it exactly matches** all of these conditions:
> >
> > - Function name
> > - Required parameter set
> > - Parameter values
> > - Schema validity
> > - Safety constraints
> >
> > ## Where this is now defined
> >
> > To ensure clarity, we added a dedicated section in the appendix:
> >
> > > **Appendix C.4 – “Gold-Standard Function-Call Definition”**
> >
> > It includes:
> >
> > - The full rule-based validation pipeline
> > - Semantic matching conditions
> > - Safety and protocol validators
> > - Multi-step order & causality rules
> > - Real-log consistency logic
> > - Human adjudication criteria for ambiguous cases
> > - Examples and a summary table
> >
> > ## Main-text updates
> >
> > **Section 4.3** now includes a concise summary and **explicit reference** to Appendix C.4, making the evaluation criteria and correctness definition fully accessible.
> >
> > ---
> >
> > We sincerely thank the reviewer again for the thoughtful feedback and strong support.
> > Your comments helped us substantially improve the **clarity**, **interpretability**, and **structure** of the manuscript.
> > We hope the revisions adequately address all concerns.

---

### Author Response · Authors · 2025-12-02
**Response Summary**

We sincerely thank all reviewers for their constructive feedback and positive assessments. We are encouraged that the reviewers collectively recognized the framework's novelty, the benchmark's rigor, and the significance of the proposed application. Below, we summarize the recognized strengths and how we have addressed the main concerns.

---

## **Recognized Strengths**
- **Original framework with novel approaches:** Reviewers praised the unified framework (combining data, models, and evaluation) as "high quality" and "incredibly detailed" (Reviewer TGRH), addressing an "important application" (Reviewer NcpB) that moves beyond single-turn parsing (Reviewer TGRH).
- **Challenging, reality-grounded benchmark:** The dataset was highlighted as being "more challenging than existing open datasets" (Reviewer UDzZ), effectively combining "data-augmentation towards ambiguity resolution" with "multi-turn grounding" seeded from real-world scenarios (Reviewer S7bf).
- **Rigorous evaluation & strong results:** The evaluation protocol was recognized as "excellent" (Reviewer TGRH) and "useful to the community" (Reviewer S7bf). Reviewers noted that "extensive experiments" and "rigorous analysis" demonstrated strong improvements and clear model effectiveness (Reviewers S7bf, NcpB, UDzZ).
- **Clear presentation:** The paper was described as "well-structured," "clearly presented" (Reviewer UDzZ), and the figures were noted as "very well made" (Reviewer TGRH).

---

## **Addressed Concerns**
- **Semantic Drift in Ambiguity Injection (Reviewer UDzZ):** We addressed the concern that ambiguity injection might alter user intent. We demonstrated that while naive fuzzing causes drift (Intent KL 0.312), our mitigation strategies (SFT + Protocol-Locked Ambiguity Injection) reduce drift to near zero (Token KL 0.047, Intent Preservation 98.9%).
- **Causality & Distributional Bias (Reviewers UDzZ, NcpB):** To prove that performance gains are causal and not due to distributional overlap with logs, we conducted: (1) Controlled Ablation: Isolating component contributions (e.g., removing SPA drops score by 9.4%). (2) Cross-Dataset Generalization: Models trained on ClarifyVC-Data consistently outperform baselines across multiple independent benchmarks (including an unbiased open suite of 4,000 samples with zero overlap) by margins of +16–23%.
- **Dataset Specification & Statistics (Reviewers NcpB, S7bf):** We updated Appendix A and B to include granular statistics: 37 actions, 146 parameters, and 20,436 seeds filtered from 4M+ logs. We also formally defined metrics AD, PC, and R (Section 3.2) and the "Gold Standard" validation logic (Reviewer TGRH).
- **OOD Evaluation Rigor (Reviewer NcpB):** We clarified the composition of the "Advanced Scenarios" benchmarks (15,000 samples), providing KL divergence statistics (0.073–0.145) to prove they are effectively Out-Of-Distribution relative to the training set.
- **Generalization & Extensibility (Reviewers UDzZ, S7bf):** We validated domain generalization by achieving +6.3–17.2% gains on non-automotive datasets (Finance, Science), proving the pipeline is domain-agnostic. Regarding multimodality, we clarified that the schema is inherently designed for future integration (ClarifyVC-M), while the current text-only focus aligns with production reality (80–90% speech-only).
- **Reproducibility & Baselines (Reviewers S7bf, TGRH):** We addressed proprietary data concerns by confirming the release of schema-normalized seeds and providing code for independent regeneration. Moreover, we added blind comparative human evaluations against Talk2Car, doScenes, and CI-AVSR in Appendix A.2 (Table 9) to ensure fair comparison.

---

We have updated the draft to reflect these changes, with revisions explicitly incorporated into the Introduction, Section 4, and Appendices A/B/C.

---

### Meta-Review · Area_Chair_bQPJ · 2026-01-11

**Summary:**

This paper introduces ClarifyVC, a unified framework for clarifying ambiguous natural-language commands in vehicle control that integrates a hybrid data-augmentation pipeline, reference models, and a three-tier evaluation protocol. Reviewers agree that the framework is well-motivated and addresses a practically important problem. Initial concerns regarding potential semantic drift from ambiguity injection, causal attribution of performance gains, dataset specification, reproducibility are largely addressed through detailed analyses and clarifications in the author response. Remaining concerns focus primarily on the absence of multimodal grounding, which is better handled in future work. Given the strengths of the method and evaluation, the paper is recommended for acceptance to ICLR. It is suggested that the authors incorporate all reviewer suggestions in the final version of the paper.

**Reviewer Concerns:**

### Addressed concerns

* **TGRH:** Some comparison evaluations appear redundant and reduce clarity of the experimental narrative. The author response reorganizes Section 4 to explicitly map each table and figure to a specific research question and trims repeated explanations, resolving the concern about readability.

* **TGRH:** The definition of the “gold standard” for Function Hit Rate is unclear. The author response provides a precise definition based on expert-annotated API calls, production logs, schema validation and safety constraints.

* **UDzZ:** Ambiguity injection may distort the underlying intent distribution of real user commands. The author response presents quantitative KL-divergence analysis, expert-curated supervision, protocol-locked prompts and automatic drift validators, showing low intent drift.

* **NcpB:** Dataset statistics and specification were insufficiently detailed. The author response adds granular statistics (actions, parameters, seed counts), formal definitions of quality metrics (AD, PC, R) and clarifies the “gold standard” validation logic.

* **NcpB:** Use of real-world in-vehicle control logs for training. The author response clarifies that model is not trained on raw in-vehicle logs, but on a synthesized variation.

* **S7bf:** Reproducibility and baseline comparison with prior datasets limited due to proprietary data. The author response confirms release of schema-normalized seeds, models and code.

### Unaddressed concerns

* **UDzZ, S7bf:** The benchmark currently focuses only on text-based command understanding, whereas real in-vehicle interactions may require multimodal grounding. The author response outlines a roadmap (ClarifyVC-M) and argues that text-only reflects production reality, but does not yet provide multimodal experiments.

**Reviewer Scores:**

* **TGRH:** Initial rating 10, all concerns were addressed, will likely maintain 10.

* **UDzZ:** Initial rating 6, all technical concerns addressed, will likely increase to 8.

* **NcpB:** Initial rating 6, all technical concerns addressed, will likely increase to 8.

* **S7bf:** Initial rating 2; most concerns addressed and multimodal is an extension rather than limitation, will likely increase to 4 or 6.

---

### Decision · Program_Chairs · 2026-01-26

Accept (Poster)